# The Quantification and Correction of Wind-Induced Precipitation Measurement Errors

John Kochendorfer[1], Roy Rasmussen[2], Mareile Wolff[3], Bruce Baker[1], Mark E. Hall[1], Tilden Meyers[1], Scott Landolt[2], Al Jachcik[2], Ketil Isaksen[3], Ragnar Brækkan[3], and  Ronald Leeper[4,5]

[1]ARL/Atmospheric Turbulence and Diffusion Division, National Oceanic and Atmospheric Association, Oak Ridge, TN, 37830, US

[2]National Centers for Atmospheric Research, Boulder, 80305, US

[3]Norwegian Meteorological Institute, Oslo, 0313, Norway

[4] N Carolina State Univ., Cooperative Inst of Climate and Satellites, Asheville, 28801 US

[5] National Center for Environmental Information, National Oceanic and Atmospheric Association, Asheville, 28801 US

*Correspondence to*: John Kochendorfer (john.kochendorfer@noaa.gov)

## Abstract

Hydrologic measurements are important for both the short and long term management of water resources. Of the terms in the hydrologic budget, precipitation is typically the most important input; however, measurements of precipitation are subject to large errors and biases. For example, an all-weather unshielded weighing precipitation gauge can collect less than 50% of the actual amount of solid precipitation when wind speeds exceed 5 m s$^{-1}$. Using results from two different precipitation testbeds, such errors have been assessed for unshielded weighing gauges and for weighing gauges employing four of the most common windshields currently in use. Functions to correct wind-induced undercatch were developed and tested. In addition, corrections for the single-Alter weighing gauge were developed using the combined results of two separate sites in Norway and the US. In general, the results indicate that the functions effectively correct the undercatch bias that affects such precipitation measurements. In addition, a single function developed for the single-Alter gauges effectively decreased the bias at both sites, with the bias at the US site improved from -12% to 0%, and the bias at the Norwegian site improved from -27% to -4%. These correction functions require only wind speed and air temperature as inputs, and were developed for use in national and local precipitation networks, hydrological monitoring, roadway and airport safety work, and climate change research. The techniques used to develop and test these transfer functions at more than one site can also be used for other more comprehensive studies, such as the World Meteorological Organization Solid Precipitation Intercomparison Experiment (WMO-SPICE).

## 1 Introduction

Precipitation measurements are used by policy makers, hydrologists, farmers, and watershed managers to quantify and allocate the water available for society's needs. Precipitation measurements are necessary for public safety in areas as

diverse as avalanche control, flood forecasting, roadway safety, and aircraft de-icing operations. Precipitation measurements are also used to evaluate radar-based estimates of rainfall, to monitor climate change, and help improve climate and weather models. More specifically, monitoring changes in the frequency, intensity, duration and phase of precipitation is critical for current and future climate research (Barnett et al., 2005; Blunden and Arndt, 2016; Trenberth, 2011; Trenberth et al., 2003).

5   Although precipitation has been monitored for many centuries, the need to accurately measure precipitation will become even more important in the future, as changes in precipitation are predicted to be complex and variable with location, requiring robust and accurate precipitation measurements (Trenberth, 2011). Despite this critical need, precipitation observations are still beset with significant biases and errors (e.g. Adam and Lettenmaier, 2003; Førland and Hanssen-Bauer, 2000; Groisman and Legates, 1994; Scaff et al., 2015; Vose et al., 2014; Yang et al., 2005).

Solid precipitation is particularly difficult to measure accurately, and biases between wintertime precipitation measurements made using different technologies, in different measurement networks, or across different regions can be larger than 50% (Mekis and Vincent, 2011; Rasmussen et al., 2012; Yang et al., 1999). Previous studies have identified wind effects as one of the primary causes for snow undercatch (e.g. Folland, 1988). This is due to two important factors: 1) the relatively slow fall

velocity of snow, and 2) the creation of flow distortions of similar magnitude to the snow fall velocity by the gauge itself as air flows past it. These two factors cause a snowflake trajectory to be significantly deflected by the airflow past the gauge. In particular, the updraft at the leading edge of the gauge can lead to an upward deflection of snowflake trajectories, causing snow to miss the gauge orifice and not be measured. The flow distortion around the gauge increases as the wind speed increases, while snowflake terminal velocity remains the same, causing more snowflakes to be deflected around the gauge.

Thus, the amount of precipitation caught in a precipitation gauge relative to the reference or actual amount of precipitation, (also referred to as the collection efficiency) decreases with increasing wind speed. The collection of rain in a weighing gauge suffers from this same problem (e.g. Duchon and Essenberg, 2001), but to a much lesser extent due to the order of magnitude higher fall velocity of raindrops, allowing them to be subject to only minimal trajectory deflection due to flow distortions around the gauge.

Adjustments, also referred to as transfer functions, are used to correct the gauge undercatch caused by the wind (e.g. Goodison, 1978). Such transfer functions are derived from precipitation testbed measurements, and describe the collection efficiency (also referred to as catch efficiency) for a specific gauge-shield system. These transfer functions determine the catch efficiency as a function of wind speed for different precipitation types (e.g. Yang et al., 1999), or more recently as a

continuous function of both wind speed and air temperatures (Wolff et al., 2015).

In addition, to mitigate the deflection and undercatch of snow by weighing gauges, wind shields are used around the gauge with the goal of slowing the oncoming flow, and lessening the resulting flow distortions around the gauge. Prominent among these is the Alter shield (Alter, 1937). This shield consists of vertical slats of ~40 cm length suspended from a circular rim

~100 cm in diameter surrounding the gauge. The flow speed is indeed decreased (Rasmussen et al., 2012), and the flow distortion around the gauge reduced, leading to an increase in snow collection. This shield is widely used around the world to improve measurements of snow and rain.

To quantify the impact of the Alter shield and other types of shields (Alter, 1937; Nipher, 1878; Yang et al., 1995), the World Meteorological Organization (WMO) sponsored a solid precipitation intercomparison study for manual precipitation gauges in the early 1990s (Goodison et al., 1998). A significant result of this study was the creation of a reference standard gauge for snowfall. This was determined to be a precipitation gauge embedded within a carefully pruned bush without leaves. The height of the bush was the same level as the orifice of the gauge. A secondary reference was established as the

Double Fence Intercomparison Reference (DFIR, Groisman et al., 1991; Yang, 2014). This allowed field sites to establish a standard for snow measurement using a DFIR without having to grow and prune a bush system. A key characteristic of the reference was that the snowfall rate did not significantly depend on the magnitude of the wind.

This earlier WMO study used both the bush gauge and DFIR as references to compare to a variety of gauges with different

shielding configurations (Goodison et al., 1998). A key result from this study was that the collection efficiency of the gauge was primarily determined by the type of wind shield used (Yang et al., 1999). Rasmussen et al. (2012) compiled a review of snowfall measurements to date that confirmed this result for automatic weighing gauges and other wind shield types. These studies confirmed that the weighing gauge collection efficiency for snow, as compared to the above-mentioned reference gauge systems, decreased to varying extent with increasing wind speed depending on the type of the shield. In addition, past

studies have shown that the collection efficiency for a given wind speed and wind shield/gauge type can vary significantly (Yang et al., 1999, Rasmussen et al., 2012). While functions describing the decrease in collection efficiency could be derived, there was often as much variability at a given wind speed as across the range of wind speeds (Yang et al., 1999). These results revealed that wind speed is not the only factor that impacts the trajectory of a snowflake past a gauge. Since the fall speed of a snowflake is often close to the magnitude of the flow distortion, an obvious candidate was the fall speed of the

snow. Using numeric modelling, Theriault et al. (2012) showed that the difference between wet and dry snow fall speeds can lead to significant changes in the collection efficiency of snow by a Geonor weighing gauge with and without an Alter shield. Other factors that impact collection efficiency include airflow turbulence, time dependence of the flow past a gauge (Colli et al., 2015), snow size distribution (Theriault et al., 2012), ice crystal habit, and snow density (Colli et al., 2015).

The performance of weighing gauges with various types of wind shields may therefore be expected to vary by climate region. The previous WMO Solid Precipitation Intercomparison (Goodison et al., 1998) examined solid precipitation from manual gauges at a limited number of field sites. The more recent WMO Solid Precipitation Intercomparison Experiment (WMO-SPICE) expanded the number of climate regimes covered, and used automatic gauges instead of manual (Nitu et al.). This paper examines results from two sites participating in this experiment, and includes several years of measurements that

pre-date the WMO experiment. These results include: 1) the assessment of various weighing gauge/shield combinations across these sites; 2) a description of the dependence of snow collection efficiency on wind speed, as well as its uncertainty, at two sites with different characteristic climatological conditions; 3) the functional form of the collection efficiency/wind speed relationship (transfer function); 4) an assessment of whether each climatological site requires a different transfer function, or if a single multi-site function can be used to reduce wind induced snow undercatch; and 5) quantification of the expected uncertainty of the correction as a function of gauge/shield type and climatological conditions.

## 2 Site descriptions

The US field site is located just outside of Boulder, Colorado, along the eastern slopes of the Colorado Front Range (Fig. 1a). The site resides on top of the Marshall Mesa at 39.949° north, 105.195° west, and is ~1740 m above sea level. Prairie grasses and low scrub are the primary flora found at the site (Fig. 1 d and e), and its proximity to the mountains makes it an ideal location for studying upslope snowfall events. Although this site experiences only intermittent snow on the ground and the snow depth is typically less than 0.5 m, it receives a total of ~ 200 cm snowfall throughout the course of the winter months. Snowfall at the site typically occurs from October through April, but can occasionally occur as early as September or as late as June. The site is relatively flat, and the lack of trees, large buildings and other obstacles allows for uninterrupted wind flow around the gauges. The site has several Double Fence Intercomparison Reference (DFIR) shields. A single-Alter (SA) shield with a weighing precipitation gauge within the DFIR was used as the reference configuration for this study. All of the gauges included in this study were mounted with their inlets approximately 1.9 m above the ground. The layout was designed to minimize the effects of the gauges and their wind shields on each other. The site generally stretches out perpendicular to the wind direction that prevails during snow storms, and special care was taken not to put larger shields upwind of unshielded and smaller-shielded gauges. Comparison of replica single- and double-Alter shielded gauges from differing locations at the field site and a lack of any significant relationship between wind direction and catch efficiency (e.g. for the single-Alter gauge, $R^2 = 0.001$) indicated that wind direction did not play an important role in catch efficiency at the site. The site is also described in detail in past publications (Rasmussen et al., 2012; Rasmussen et al., 1999).

The Norwegian Haukeliseter test site (abbreviated as NOR) is situated at 59.812° N, 7.214° E, and is at 991 m a.s.l. on a plateau in an alpine region in southwestern Norway (Fig. 1a). Snowfall at the site typically occurs October through May. Annual snow depth often reaches up to 3 m, although during the time period included in the present study, it only reached 1.97 m. The site has one DFIR shield surrounding an automated weighing gauge within a single-Alter shield. The DFIR gauge and several other gauges, most of them within single-Alter shields, were installed in two lines perpendicular to the main wind directions (east-southeast and west-northwest). An evaluation of two similar sets of precipitation and wind measurements recorded at different locations at the site during February - May 2010 indicated that the site was homogeneous (Wolff et al., 2010), with comparison of the two sets of measurements resulting in a correlation coefficient of 0.94, a

standard deviation of 0.07 mm (8.2%), and similar average event precipitation from both locations (0.89 mm and 0.87 mm). The gauges at the Norwegian test site were all mounted 4.5 m above ground level in order to mitigate the effects of blowing snow and to allow for the increasing snow depth through the season. The site is described in more detail in previous publications (Wolff et al., 2013; Wolff et al., 2015).

## 3.0 Methods

### 3.1 Measurement period

The US measurements used here span from Jan 01, 2009, through March 07, 2014, and include all seasons. The NOR measurements used here are identical to those used in Wolff et al. (2013), and were recorded only during winter periods as follows: Feb 01, 2011 – Apr 30, 2011; Nov 01, 2011 – Apr 30, 2012, and Feb 01, 2013 – May 31, 2013.

### 3.2 Precipitation gauges and shields

To reduce potential sources of uncertainty, all of the precipitation measurements presented here were recorded using the same model weighing precipitation gauge (3-wire T200B, Geonor Inc., Oslo, Norway), although there were differences in the Geonor gauge capacities, with 1000 mm gauges used at the NOR site and 600 mm gauges used at the US site. The gauge inlets were all heated using the same type of inlet heaters (described in NOAA Technical Note NCDC No. USCRN-04-01), with both the upper (exterior) and lower (interior) sections of the inlet heated to prevent snow melted at the orifice from re-freezing when it drips into the collection bucket. The inlet heaters were activated only when the inlet temperature and the air temperature were both < 2 °C. At the US site, 2 litres of antifreeze (60% Methanol, 40% Propylene Glycol) and 0.4 litres of hydraulic oil (Lubriplate Minus 70) were added to every gauge to prevent freezing and evaporation. At the NOR site, 5 litres of Methanol, 3.3 litres of Ethylene Glycol, and 0.4 litres of hydraulic oil (Hydraway HVXA 15LT) were used.

Present at both the NOR and US sites, the DFIR shield has the largest footprint of any of the shields, and consists of three concentric shields. The outer two shields are octagonal in design, and are made out of 1.5 m tall wood laths, with an outer shield diameter of twelve meters and a middle shield diameter of four meters. The DFIR shield has a porosity of 50%, with 50% of its surface area open, allowing air to pass though (the other 50% of its surface area is blocked by the wood laths), and both the outer and middle shields are perpendicular to the ground. For the third innermost shield, an Alter-style shield of standard size and configuration is used. The DFIR shield is described in more detail in the first WMO Solid Precipitation Intercomparison (Goodison et al., 1998).

The 2/3 scale version of the DFIR, hereafter referred to as the small DFIR (SDFIR), was designed for the US Climate Reference Network program and was installed only at the US site. The SDFIR laths are 1.2 meters long, the diameter of the outer shield is eight meters in diameter, and the middle shield is 2.6 meters in diameter. Additionally, the middle shield

height is 10 cm lower than the outer shield. A standard diameter Alter shield is used as the innermost shield, which is 10 cm lower than the middle shield and located at the same height as the gauge orifice.

Single-Alter shields were installed at both the NOR (foreground of Fig. 1b) and US sites. The single-Alter shield consists of metal laths about 40 centimeters in length (though some versions of the Alter use slightly longer laths that are 46 centimeters in length). The laths on the SA shield are typically attached near the top to a circular ring, 1.2 meters in diameter, and allowed to move freely in the wind. The double-Alter shield (DA, Fig. 1d) is a variation of the single-Alter shield and has two concentric shields instead of one (Rasmussen et al., 2001). This shield consists of a standard 1.2 meter diameter single-Alter shield surrounded by an additional outer ring of laths measuring two meters in diameter. Like the single-Alter, the laths on both rings are approximately 40 cm in length, and secured only at the top, allowing them to move freely at the bottom. Drawings of the DFIR, single-Alter shields, and double-Alter shields, and descriptions of their effects on the wind speed at the gauge inlet are available in Rasmussen et al. (2012).

The Belfort double-Alter shield was only present at the US site (Fig. 1e), and is a modified version of the standard double-Alter shield. The diameter of the inner shield is 1.2 meters and the laths are 46 cm long. The diameter of the outer shield is 2.4 meters and the laths are 61 cm long. Unlike the standard single and double-Alter shield laths, these laths don't taper at the bottom and are only allowed to swing inwards or outwards at a maximum 45-degree angle. The Belfort double-Alter shield is also only approximately 30% porous, which is significantly less than the ~50% porous double-Alter shield.

Unshielded gauges (e.g. Fig. 1c) were present at both sites, but due to problems with the unshielded measurements from the NOR site, only the unshielded measurements from the US site were used for the development and testing of transfer functions.

### 3.3 Other measurements

### 3.3.1 US Site

At the US testbed, the air temperature was measured using fan-aspirated (Model 076B Radiation Shield, Met One Instruments, Grants Pass, OR, US) platinum resistance thermometers (Thermometrics, Northridge, CA, US) mounted at a height of 1.5 m. Three wetness sensors (Model DRD11A, Vaisala, Helsinki, Norway), also mounted at a height of 1.5 m, were used to independently detect precipitation. Wind speed was measured at 1.5 m using a cup anemometer (Model 014A Wind Speed Sensor, Met One Instruments, Grants Pass, OR, US), at 2 and 3 m using propeller anemometers (Model 05103 Wind Monitor, RM Young, Traverse City, MI), and at 10 m using both a propeller anemometer (Model 05103 Wind Monitor, RM Young) and a two-dimensional sonic anemometer (Model 86004 Ultrasonic Anemometer, RM Young). The two anemometers at 10 m were found to interfere with each other due to wind shadowing when winds were from the north or

the south, and a composite 10 m wind speed was therefore produced using the ultrasonic anemometer measurements to replace the propeller anemometer measurements when winds were from the north (wind directions < 30° or > 340°). Likewise, as identified by plotting the ratio of the measured wind speed to the 10 m wind speed as a function of wind direction, the lower wind speeds were found to be subject to interference at some wind directions. Using 30-min mean wind speeds measured in a clear sector (wind direction < 30°) from all measurement locations, the roughness length ($z_0$ = 0.01 m) and displacement height ($d$ = 0.4 m) were determined based on the log wind profile (Thom, 1975):

$$U_z \approx \ln\left[(z - d)/z_0\right] \tag{1}$$

where $U_z$ is the wind speed ($U$) at a height $z$ ($U_z$). Using the same relationship, the wind speed at the gauge height of 1.9 m was estimated be equal to $U_{10m} \times 0.71$. Only wind speed measurements recorded during precipitation events were used to develop this relationship, minimizing the neglected effects of stability on the wind profile in the typically overcast and near-neutral surface layer conditions associated with precipitation. To mitigate the effects of near-field obstructions on the near-surface wind speed measurements, the 10 m wind speed measurement was used to estimate the gauge-height wind speed measurement using this method throughout the study. Errors in the method used to estimate the gauge height wind speed from the 10 m wind speed were evaluated using the mean half-hour 2 m wind speed measurements recorded during precipitation events when the recorded wind speed was greater than 1 m/s from unobstructed wind directions (wind direction < 30 deg). The 2 m wind speeds were compared to the gauge height (1.9 m) wind speed estimated using the log profile, resulting in a root mean square error (RMSE) of 0.4 ms⁻¹ (10.1%) and a bias of -0.12 ms⁻¹ (-3.1%).

### 3.3.2 Norwegian site

At the Norwegian site, air temperature was measured with a 100 Ω platinum resistance thermometer within a standard Norwegian radiation screen installed at gauge height on a tower near the DFIR. A precipitation detector (Precipitation Monitor, Thies Clima, Göttingen, Germany) was also mounted at gauge height and used to record the presence/absence of precipitation. The primary wind speed was measured at a height of 10 m with a heated two-dimensional sonic anemometer (WindObserver II, Gill Instruments, Lymington, UK). In addition, a propeller anemometer (Wind Monitor, R.M. Young, Traverse City, US) was mounted at gauge height, but was found to be obstructed by a neighbouring single-Alter shield from some wind directions. To mitigate the effects of these compromised gauge height wind speed measurements, a relationship was developed to determine the gauge height wind speed using the 10 m height wind speed. The ratio was developed using unobstructed (wind direction > 240°) 10 m and gauge height wind speed measurements recorded during precipitation events. The resulting relationship was: $U_{4.5m} = 0.93 \times U_{10m}$, $R^2$ = 0.99, RMSE = 0.54 m s⁻¹. Using this relationship, the wind speed at 10 m was used to predict the gauge-height wind speed for all wind directions. Following Wolff et al. (2015), precipitation events were also screened for wind directions associated with shadowing between gauges and shields (wind direction < 240° and wind direction > 355°), and were excluded from the analysis.

## 3.4 Precipitation type

Transfer functions have commonly been developed separately for snow, mixed precipitation, and rain (e.g. Goodison et al., 1998; Yang et al., 2005). Another proposed classification scheme involves differentiating between wet and dry snow. In the past, manual observations of precipitation type were recorded and used to develop such transfer functions, but modern automated measurement networks now rarely include such observations. Airport weather stations often include automated precipitation type measurements, but hydrological, meteorological, and climate stations do not typically include precipitation type measurements, and defensible methods to correct wind-induced errors without precipitation-type measurements are therefore needed.

At the US testbed, precipitation type was determined using an unshielded present weather detector (Vaisala PWD22, Helsinki, Finland). Half hour increments of rain, mixed precipitation, and snow were identified when more than 15 min of any of these precipitation types was detected from the minutely present weather detector measurements. For rain and snow classifications, less than 5 min total of any other precipitation type were allowed to occur.

At the US site, when the temperature was below -2.5 °C, more than 95% of the precipitation in every 1 °C bin was classified as snow using the present weather detector (Fig. 2). Above 2.5 °C, more than 95% of the precipitation was classified as rain. These thresholds will change depending on the climate of a given site, and the 95% threshold could also be adjusted to suit the needs of a given study. However, based on these temperature thresholds, the precipitation type that was most sensitive to classification methods was mixed precipitation. The present weather detector classified only 5% of the available half hours of precipitation as mixed, whereas 19% of the precipitation occurred between -2.5 °C and 2.5 °C. A significant amount of rain and snow may therefore be misclassified as mixed precipitation when temperature thresholds are used to predict precipitation type, assuming that the present weather detector accurately identified precipitation type. Wolff et al. (2015) performed a similar analysis of the precipitation type at the NOR site, using their Present Weather Detector (Vaisala PWD21, Helsinki, Finland).

An alternative to using temperature thresholds to differentiate between different precipitation types is to use a continuous function of both air temperature and wind speed to describe the catch efficiency (Wolff et al., 2015). Although the functions produced using such methods are more complex than a relationship between wind speed and catch efficiency for a single precipitation type, such an approach is arguably more convenient because only one equation is needed to determine catch efficiency for all conditions. More importantly, a continuous function of temperature may more accurately represent reality, as catch efficiency varies continuously with air temperature, especially near 0 °C. The transition from liquid precipitation to dry snow is continuous, without any well-defined step changes. In addition, a continuous function of air temperature and

wind speed eases the comparison of catch efficiency results from different sites with potentially different climates. This in turn facilitates the evaluation of the uncertainty inherent in a transfer function that describes more than one site.

Following Wolff et al. (2015), the transfer functions developed here use a continuous $f(T_{air}, U)$, as air temperature measurements are more universally available than precipitation type, different precipitation type detectors do not always agree on the precipitation type (Merenti-Valimaki et al., 2001; Sheppard and Joe, 2000; Wong), and air temperature thresholds used to separate different precipitation types must be chosen somewhat arbitrarily.

## 3.5 Transfer function development

### 3.5.1 Data analysis and event selection

For the US results, the US Climate Reference Network precipitation algorithm was used to determine 5-minute accumulations from all of the 3-wire Geonor gauges (Leeper et al., 2015). The algorithm relied upon wetness sensor measurements to detect periods of precipitation, and it calculated the average accumulation of the three wires by inversely weighting the individual wire accumulations using the variance of the individual depths over the last 3 hours. This was done to lessen the contribution of noisier wires to the total gauge depth, and thereby decrease the amount of noise in the precipitation measurements. The algorithm was modified for the purposes of this study by increasing the 5-minute precipitation resolution from 0.1 mm to 0.01 mm. The 5-minute accumulations were then summed into half-hour periods, with these half-hour precipitation measurements used for the subsequent analysis.

In addition, at the US site, more than 10 minutes total of any type of precipitation as identified by the present weather detector had to occur within each half-hour to be included in the transfer function analyses. Half-hour increments with unrealistic air temperatures or wind speeds were also excluded from the transfer function analyses. For example, half-hour increments with 10 m anemometer measurements affected by ice accumulation were identified by comparing the 10 m propeller anemometer measurements with the 1.5 m cup anemometer measurements.

For the NOR results, the methods presented in Wolff et al. (2015) were used to calculate precipitation accumulation and select 60-min precipitation periods for analysis. Every minute, gauge depths were determined by averaging the output of the three wires of each sensor. As an additional noise filter, 10 min running averages were calculated for the precipitation gauges and the optical precipitation detector. For the next step, periods with continuous and clear precipitation signals were selected using the following criteria:

1) The precipitation detector signalled precipitation for at least eight out of ten minutes;
2) The accumulation was greater than 0.1 mm per 10 minutes or greater than 1 mm for events that lasted longer than 100 min.

The resulting precipitation periods were of different lengths, and were divided into sets of 10 min and 60 min events for further analysis.

### 3.5.2 Selection of precipitation threshold

For the US site, 30-minutes was selected as the most suitable time interval for the creation of transfer functions, as 30-minute averages of air temperature and wind speed are both representative of the field site as a whole and also relatively stationary (e.g. Stull, 1988). A longer time period would be subject to increased mesoscale, synoptic, and diurnal changes in precipitation type, wind speed, and air temperature. Under quiescent conditions, a shorter averaging period would approach turbulent time scales, where the presence or absence of an individual eddy would affect the results, making the results less representative of the entire site. At the NOR site, 60-minute periods of precipitation were used following Wolff et al. (2015), with many of the same arguments supporting the 30-minute period equally valid for 60-minutes. Qualitative analyses by Wolff et al. (2015) on 10- and 60-minute datasets did not reveal any significant differences between those time intervals. Therefore the 60-minute time period, which is similar to the operational measurement frequency in Norway, was chosen for further analysis by Wolff et al. (2015).

Catch errors are described well using catch efficiency, described as the ratio between precipitation accumulated in a gauge under test and the standard precipitation accumulation ($CE = {P_{UT}}/{P_{DFIR}}$, where $CE$ is catch efficiency, $P_{UT}$ is the accumulation of precipitation from a gauge under test, and $P_{DFIR}$ is the accumulated DFIR precipitation used as the standard). Because of this, a minimum threshold is necessary to constrain errors in the denominator of the catch efficiency ratio. Using all of the 30-minute periods of snow measured within the SDFIR and DFIR gauges from the US site, a minimum 30-minute precipitation amount of 0.25 mm in the DFIR gauge was found to provide a good balance between reducing the effects of measurement noise while simultaneously maintaining a large sample size of events. This was examined by iteratively increasing the DFIR precipitation threshold from zero in 0.01 mm steps, and calculating a simple linear transfer function for each threshold. The number of 30-minute events ($n$) and standard deviation ($\sigma$) of the $CE$ model were estimated for every 0.01 mm increase in threshold, and a minimum in the standard error ($SE = {\sigma}/{\sqrt{n}}$) was encountered at 0.25 mm (Fig. 3).

For this threshold test, the SDFIR gauge data were used to develop transfer functions, as they were the most comparable to the DFIR gauge data (Table 1) and were described well by a simple linear transfer function for snow events. In reality, some variability occurs in this 'ideal' threshold based on the specific gauge/shield under test, the sample size, precipitation type, and the amount of noise in the measurements. However, for the sake of simplicity and consistency, the same 0.25 mm reference threshold was used for all of the precipitation measurements included in this study.

Using the original 30-minute SDFIR precipitation measurements, which were typically nearly equal to the DFIR gauge measurements, it was found that a minimum threshold for the gauge under test was also required to help select representative

precipitation measurements, rather than a biased sub-selection of the measurements. This was because both the gauge under test and the standard DFIR gauge were affected by random measurement error and random spatial variability in precipitation. In other words, the DFIR gauge is capable of measuring more than 0.25 mm of precipitation in a 30-minute period, even when the actual site-average rate is below this threshold. Because many solid precipitation events occur near the 0.25 mm threshold, many such events may be included in the analysis. Without the use of a minimum threshold for a gauge under test, measured catch efficiencies from a gauge under test may therefore be biased erroneously low (as a result of the DFIR gauge being biased erroneously high), and the resultant transfer function may over-correct the gauge under test. The minimum thresholds for the gauges under test were determined from Eq. 2, using the entire multi-year datasets available.

$$THOLD_{UT} = \frac{median(P_{UT})}{median(P_{DFIR})} 0.25 \; mm \tag{2}$$

Where $THOLD_{UT}$ is the threshold of the gauge under test, $P_{UT}$ is the 30-minute accumulation from the gauge under test, and $P_{DFIR}$ is the 30-minute DFIR accumulation. The resultant thresholds were 0.18 mm for the unshielded gauge, 0.20 mm for the single-Alter gauges, 0.21 mm for the double-Alter gauges, 0.22 mm for the Belfort double-Alter gauge, and 0.25 mm for the SDFIR gauge.

### 3.5.3 Choice of transfer function model

Wolff et al. (2015) tested many sigmoidal type transfer functions for the determination of catch efficiency from a single-Alter gauge as a function of $T_{air}$ and $U$. The equation they selected was sigmoidal in respect to its response to both $T_{air}$ and $U$. It is included here for reference, as it is used throughout this study:

$$CE = \left[1 - \tau_1 - (\tau_2 - \tau_1)\frac{e^{\left(\frac{T_{air}-T_\tau}{s_\tau}\right)}}{1+e^{\left(\frac{T_{air}-T_\tau}{s_\tau}\right)}} e^{-\left(\frac{U}{\theta}\right)^\beta}\right] + \tau_1 + (\tau_2 - \tau_1)\frac{e^{\left(\frac{T_{air}-T_\tau}{s_\tau}\right)}}{1+e^{\left(\frac{T_{air}-T_\tau}{s_\tau}\right)}}, \tag{3}$$

where $T_{air}$ is the air temperature, $U$ is the wind speed, and $\tau_1, \tau_2, T_\tau, s_\tau, \theta, \beta,$ , are coefficients fit to the data described in more detail in Wolff et al (2015). An example of the sigmoid function fit to SA $CE$ measurements from both NOR and US is shown in Fig. 4.

For the sake of simplicity, an alternative function is proposed, with an exponential response to wind speed, and with a simple sigmoid (tan$^{-1}$) response to $T_{air}$:

$$CE = e^{-a(U)\left(1 - [\tan^{-1}(b(T_{air}))+c]\right)}, \tag{4}$$

where $a$, $b$, and $c$ are coefficients fit to the data. The form of both of these functions follows the same form presented by others (e.g. Goodison, 1978; Wolff et al., 2015; Yang et al., 1999), with catch efficiency rather than correction factor determined, such that the inverse ($CE^{-1}$) must be used to correct actual precipitation data ($P_{DFIR} = P_{UT}/CE$ ). Comparisons of corrections based on correction factors ($P_{DFIR}/P_{UT}$) and catch efficiency ($P_{UT}/P_{DFIR}$) revealed no significant differences, so $CE$ is used in the present study because it enables comparison with past experiments. However, it is difficult to interpret $CE$

error estimates (e.g. Wolff et al., 2015), so special care has been taken in the present study to estimate transfer function uncertainties and biases that are relevant to the measurement of precipitation, rather than the measurement of *CE*.

## 3.6 Uncertainty assessment

### 3.6.1 Measurement uncertainty

The repeatability or random error of the precipitation measurements used to develop the transfer functions was estimated using four different sets of replicate gauge-shield combinations at the US and NOR testbeds. This analysis included both random gauge measurement uncertainty and uncertainty caused by the spatial variability in precipitation occurring across a field site. At the US site, there were two SA-shielded gauges and two DA-shielded gauges recording precipitation measurements throughout the field study. There was also one DFIR gauge and one SDFIR gauge, which were similar enough

in their catch to be considered identical for the purposes of estimating measurement uncertainty. At the NOR site, there were two pairs of near-identical SA-shielded gauges, but the two gauges were on opposite sides of the DFIR. This limited the amount of data available for comparison between them, as additional screening for wind directions was necessary. In addition, two different heating systems were used on the NOR gauges, with the primary gauge used for the transfer function analysis configured with the NOAA Climate Reference Network heating system (NOAA Technical Note NCDC No.

USCRN-04-01), and the secondary SA gauge using the standard Geonor heater system. Figure 5 shows the comparison results for these four sets of identical or near-identical gauge-shield combinations. The RMSE values were calculated from differences between the identical gauges, and were < 0.15 mm for all gauge-shield combinations. These results include only snowfall; the precipitation type at the US site was determined using the present weather detector to classify 30-min periods with more than 15 min of snow and less than 5 min of other precipitation types as snow, while the precipitation type at the

NOR site was determined using the air temperature, with snowfall identified when mean $T_{air} < -2$ °C.

### 3.6.2 Uncertainty of transfer functions

Uncertainty in the transfer functions was estimated by applying the transfer functions to the different gauges under test and then comparing the results to the standard DFIR precipitation measurements. To maintain some independence between the data used to develop the transfer function and the data used to test the transfer function, the uncertainty of the transfer

functions was estimated using a 10-fold cross-validation. This model validation technique randomly separated the available measurements of a given shield type into 10 equally-sized groups. The transfer function was determined using 90% of the data (9 groups), then tested on the remaining 10% of the data (1 group). This process was repeated for each permutation of the 10 groups. The coefficients describing the transfer function were based on the entire data set, but the uncertainty estimates were based on this 10-fold cross-validation. The uncertainty estimated from 10-fold cross validation and the

uncertainty estimated by circularly developing and testing the transfer function on identical data were very similar, but the 10-fold validation was used where possible for more defensible estimates of the transfer function uncertainty.

For the SDFIR-shielded gauge under test, it proved impossible to constrain the sigmoid function for all 10 iterations of the model fitting, so the cross-validation was not used and the uncertainty and the transfer function were circularly determined using exactly the same data. To assess the uncertainty of the two-site transfer function developed for the combined SA NOR and SA US measurements presented in Section 4.2, the transfer function was applied to the entire dataset using the 10-fold cross validation, and it was also applied to the two sites individually. The site-specific results were calculated to assess to what degree a single transfer function was valid for the two individual sites. Because the dataset used to create the transfer function was split into two groups for this two-site validation, 10-fold cross-validation was not used.

## 4 Results

### 4.1 Shielding errors and biases

Errors in uncorrected precipitation measurements were estimated by comparing the DFIR-shielded precipitation measurements to the other shielded and unshielded measurements. Based on the differences between the 60-min DFIR-shielded accumulation and the SA-shielded accumulation at the NOR site, RMSE values and biases were calculated (Table 1, NOR SA). The RMSE of the NOR SA measurements was 0.64 mm (51.6%) and the bias was -0.34 mm (27.1%), indicating that errors in the uncorrected data were significant. At the US site, the 30-min precipitation measurements were similarly used to calculate RMSE values and biases (Table 1), with the RMSE values and biases generally decreasing in absolute magnitude as the size or efficacy of the shield increased. For example the absolute magnitudes of the unshielded gauge (US UN) RMSE (0.30 mm or 28.6%) and bias (-0.17 mm or -16.2%) at the US site were much larger than the SDFIR (US SDFIR) RMSE (0.14 mm or 14.7%) and bias (-0.03 mm or -3.6%). The RMSE and bias for the combined SA dataset (All SA) that included US and NOR measurements fell between the US SA and the NOR SA results. In addition to quantifying the errors associated with the different shields at the sites, these uncorrected results also provide some perspective for the corrections that were developed and applied to the precipitation measurements. RMSE and bias estimates reported in both mm and percent demonstrate that a relatively small error or bias reported in mm can actually be quite significant in terms of percent. This is due to the fact that many of the 30 or 60 minute precipitation measurements were of accumulations less than 0.5 mm, particularly for snow.

### 4.2 Transfer functions and uncertainty

For the US site, precipitation catch efficiencies were described as a function of $T_{air}$ and $U$ using both the sigmoid function (Eq. 3) and exponential function (Eq. 4) for Geonor T-200B precipitation gauges in unshielded (UN), single-Alter (SA), double-Alter (DA), Belfort double-Alter (BDA), and small DFIR (SDFIR) configurations. These transfer functions were developed for both the gauge height wind speed (Table 2) and the 10 m height wind speeds (Table 3). In addition to creating transfer functions for the individual NOR and US site SA measurements, the single-Alter (SA) results from the NOR and US

sites were combined and used to create two-site exponential (Exp) and sigmoid (Sig) transfer functions (labelled as 'All SA' in Table 2 and 3).

The RMSE and bias in the corrected measurements were then determined for each transfer function using both the gauge height wind speeds (Tables 4) and the 10 m height wind speeds (Table 5). This includes the two-site SA transfer functions and the associated errors determined from the combined dataset ('All SA' in Table 4 and 5). In addition, by applying the All SA transfer functions individually to the NOR and US SA measurements, RMSE values and biases were estimated separately for the US and NOR SA measurements using the two-site transfer functions (Tables 6 and 7). This was done to evaluate site-biases and the effects of different climates on the transfer functions. For example, for the SA gauges at the US site, the two-site gauge height wind speed Exp transfer function reduced the RMSE from 23.6% (Table 1) to 15.4% (Table 6) and improved the bias from -11.7% (Table 1) to -0.2% (Table 6). Likewise, the NOR SA RMSE was decreased from 51.6% (Table 1) to 36.4% (Table 6) and the bias was improved from -27.1% (Table 1) to -4.1% (Table 6). This indicates that the two-site transfer function effectively reduced the bias at both sites. The NOR SA and US SA transfer functions determined separately for the two sites did not perform significantly better at the individual sites (Table 4 and 5) than the two-site transfer function (Table 6 and 7), with the only notable improvements from the individual site transfer functions in the NOR biases. For example, using the Exp function for the gauge height wind speeds, the individual site transfer function improved the bias to -1.1% (Table 4), whereas the two-site transfer function only improved it to -4.1% (Table 6).

The RMSE values of the corrected results reflect the significant residual variability in the corrected catch efficiency. It is worth noting that the RMSE of even the corrected SDFIR measurements was greater than 0.1 mm, indicating that such uncorrectable errors may be due to random measurement error and site variability rather than crystal type and wind speed effects. That is, even a gauge that is shielded quite similarly to the reference, which likely responds to wind speed and crystal habit similarly to the DFIR, was subject to such errors; in a given half-hour period the SDFIR and DFIR shields are subject to similar hydrometeors and wind speeds, and these hydrometeors presumably behave the same way over each shield. Such uncorrectable errors can therefore be attributed to more random causes, such as measurement noise and the natural spatial variability of precipitation.

Generally, differences between the Exp and Sig functions were small, indicating that the Exp function can be used as a simpler alternative to the Sig function developed by Wolff et al. (2015). For the US site, the transfer function RMSE values were about 0.15 mm (15%) for all gauges, irrespective of whether the sigmoid function or the exponential function was used (Table 4 and 5). The errors in the uncorrected data (Table 1) were generally much larger than the errors in the corrected data, with the errors in the uncorrected data dependent upon the efficacy of the shield. For example, using the gauge height wind speed transfer functions, the corrected SDFIR gauge RMSE and bias shown in Table 4 were only slightly better than the

uncorrected RMSE and bias shown in Table 1, whereas application of the transfer function resulted in a much more significant improvement in the unshielded and SA gauge error (UN and SA, in Tables 1, 4 and 5).

The RMSE values were much larger for the NOR SA measurements than the US SA measurements. This was due primarily to a generally more noisy gauge at the NOR site than at the US site. Random measurement error from these vibrating-wire weighing gauges can be reduced via trial and error by rotating and remounting the vibrating wires within the gauge and also by mounting the shield separately from the gauge, but such noise can vary significantly from gauge to gauge and even from wire to wire within a gauge equipped with redundant vibrating wires. The NOR site is also windier than the US site and the undercatch is generally larger, and because of this, one can expect the RMSE to increase (i.e. a well-shielded gauge that requires less correction will be less affected by variability in precipitation type and crystal habit). In addition, blowing snow may have increased the RMSE values of the NOR results, with 2.6% of the events occurring wind speeds greater than 15 ms$^{-1}$.

The bias found for the NOR SA gauge corrected using the two-site transfer function is, however, more notable, as the bias should be relatively unaffected by random measurement noise. For example, the NOR SA measurements corrected using the gauge-height transfer function had a larger bias (approximately -0.05 mm or -4% as the mean of the Exp and Sig function results) than the US SA gauge (0.00 mm or ~0%). This indicates that small site-biases exist that can affect multi-site transfer functions, and such a bias has been quantified here using automated gauges for the first time.

For the gauges at the US site, there was no difference in the RMSE or bias between the 10 m wind speed and the gauge-height wind speed transfer functions. However, this is neither surprising nor noteworthy, as the gauge-height wind speed was estimated based on the 10 m wind speed. It is notable, however, that the RMSE for the combined US and NOR SA gauge results was also not significantly affected by the choice of wind speed measurement height (Table 6 and 7). These RMSE values indicate that although the gauge heights at the US site and the NOR site were significantly different, there was no significant loss in accuracy when transfer functions were created for both sites using the 10 m wind speed. There was, however, a more negative bias in the 10 m wind speed NOR SA results (e.g. Sig bias = -0.10 mm, or -7.8%) than for the gauge height wind speed NOR SA results (e.g. Sig bias = -0.06 mm, or -5.2%), indicating that there may be a small advantage to using the gauge height wind speed in preference to the 10 m wind speed for the development and application of precipitation transfer functions.

To demonstrate both the importance and the limitations of the transfer function corrections, the 30-min uncorrected (Fig 6a) and corrected (Fig 6b) SA snow ($T_{air} < -2.5$ °C) measurements were compared to the DFIR precipitation measurements. The uncorrected SA snow measurements are subject to significant errors as a result of variability in wind speed and crystal type, with dense, wet, warm snow and low-wind speed snow less affected by shielding than cold, dry, light snow and windy snow.

In addition, the corrected SA snow measurements reveal the effects of gauge noise, the spatial variability of precipitation, and also variability in crystal habit that are inadequately captured by air temperature. For example, at a given temperature, variability in crystal type has a significant effect on hydrometeor fall velocity, drag coefficient, and the resultant relationship between wind speed and $CE$ ( Colli et al., 2015; Theriault et al., 2012).

To demonstrate further the necessity of the transfer functions and the effects of errors and variability in the transfer functions, an example event is shown Fig. 7a. One minute accumulations during this event were corrected using the appropriate transfer function and the mean 1-min $T_{air}$ and $U$ (Fig. 7 b). This 'typical' event was, in fact, somewhat atypical, with $CE$ lower than predicted by the transfer functions. This is another example of the results shown in Fig. 6b, with some periods significantly overcorrected and others significantly under-corrected by transfer functions. These types of events serve as a good example of why it is always preferable to make the most accurate measurement possible, and only rely upon corrections when absolutely necessary. In comparison with the uncorrected precipitation values shown in Fig. 7a however, Fig. 7b also demonstrates the necessity of such transfer functions. For example, with a standard DFIR accumulation of 22.5 mm for the entire event, the UN accumulation was improved from 8.5 mm (38% of the DFIR) to 17.8 mm (79% of the DFIR).

## 5 Discussion

### 5.1 Wind speed for transfer functions

Wind speed measurements at gauge-height and at the standard 10 m measurement height both have advantages. The 10 m height wind speed is widely used by national weather services, is designated as the standard wind speed measurement height by the WMO, and is also less likely to be affected by obstacles such as towers and precipitation gauge shields. The effects of obstacles at gauge height were apparent in both the NOR and US sites, for example (Sections 3.3.1 and 3.3.2). However, the catch efficiency of a shielded or unshielded gauge is more closely linked to the wind speed at gauge height. If, for example, the wind speed at gauge height is affected by the changing height of the snowpack or by vegetation or other obstacles, this will affect the relationship between the 10 m and gauge height wind speed, and potentially lead to additional sources of error.

For the present study, in which the 10 m wind speed and a gauge height wind speed derived from the 10 m wind speed were both used to create two-site transfer functions for single-Alter shielded gauges at significantly different heights, the RMSE of the combined NOR and US transfer function was relatively insensitive to the different wind speed heights when tested on the US, NOR (Tables 6 and 7), or combined datasets (All SA, in Tables 4 and 5). The only notable change caused by the use of the 10 m wind speed versions of the combined transfer function was in the bias calculated from the NOR SA, which was about -4% (-0.05 mm) for the gauge height transfer functions (Table 6) and -7% (-0.09 mm) for the 10 m wind speed transfer functions (Table 7).

However, based on the argument that catch efficiency is physically more closely tied to the wind speed at the height of the gauge than at a height of 10 m, the correction of precipitation measurements using a gauge height wind speed measurement, or an approximation of the gauge height wind speed, is more defensible than the use of a 10 m height wind speed. Because

differences in gauge height are common due to the necessity of mounting gauges and shields well above the highest expected snow depth, even when only 10 m wind speed measurements are available they should be adjusted to gauge height for the application of shielding corrections. For example, at a gauge height of 5 m, the wind speed affecting the catch efficiency of the gauge would typically be ~90% of the 10 m wind speed, while the wind speed affecting a gauge at 2 m would be ~70% of the 10 m wind speed. Using the log wind profile and/or other available wind speed profile measurements as demonstrated

here, the commonly available 10 m wind speed can be used to estimate gauge height wind speeds and defensibly correct wind speed precipitation errors at all gauge heights. This approach has the advantage of being based on the arguably easier to measure and more commonly available 10 m height wind speed, but it also suffers from the disadvantage of being only an estimate of the gauge height wind speed. Due to obstacles at both the NOR and US site affecting the gauge height wind speeds, the gauge height and 10 m transfer functions were both by necessity derived from the 10 m height wind speeds, and

the measurements therefore cannot easily be used to compare true gauge height wind speed and derived gauge height wind speed transfer functions. It is nevertheless interesting that a two-site 10 m wind speed transfer function performed almost as well as the estimated gauge height wind speed transfer function, especially considering the large difference between the gauge heights at the two sites. It is also worth noting that at the NOR site, where snow depth during this measurement campaign had a maximum of almost 2 m, changing snow depth did not significantly affect the relationship between the 10 m

wind speed and the gauge height wind speed. Based on this, it appears that despite the theoretical advantages of using gauge height wind speeds (or estimated gauge height wind speeds) for the correction of precipitation measurements, in practice, the best available wind speed measurement will vary by site and by network, and in some cases the most representative wind speed measurement may not be at gauge height.

## 5.2 A two-site transfer function

Uncertainties and biases associated with the development and application of a single transfer function for two separate sites within differing climate regions have been presented. The NOR site was much windier than the US site; during precipitation events the mean NOR $U_{10\,m} = 8.74$ m s$^{-1}$, and the mean US $U_{10\,m} = 4.52$ m s$^{-1}$. The NOR measurements were also generally noisier than the US results; for corrected SA events with similar wind speeds ($U_{10\,m} \leq 10$ m s$^{-1}$), the NOR RMSE was 0.38 mm and the US RMSE was 0.15 mm. It is therefore difficult to draw site-specific conclusions based on the RMSE values, as

some of the gauge noise from NOR appears to be related to the specific installation, rather than the climate. However, the bias found for the NOR SA results was more significant than for the US SA measurements. This suggests that small biases between different sites and climates indeed exist, and the application of a multi-site transfer function based on wind speed and air temperature will be subject to such variability. The results from the WMO Solid Precipitation Intercomparison

Experiment will provide a better opportunity for quantifying such climate biases, as measurements from the two sites in this study will be included along with other sites in varying climates. Additionally, WMO-SPICE data from all sites will be processed using a common approach developed within the WMO-SPICE project (Reverdin).

Such a multi-site transfer function is needed, because site-specific transfer functions can only be developed for sites where a DFIR shield is present. A more broadly applicable transfer function is necessary for real-world precipitation corrections, where the actual or DFIR-shielded amount of precipitation is unknown. In this study, the bias was shown to be minimized at sites in two separate climates using one transfer function that was developed from combined results, and we suggest using this same approach in future studies such as the WMO Solid Precipitation Intercomparison, for which many more sites and
climates will be included.

## 5.3  The sensitivity of site biases to differences in analysis methods

Measurements presented in this study were recorded at two separate sites. Although similar methods were used to analyse the results from each site, the datasets available from each site were not developed exactly the same way. The methods used to determine when precipitation occurred were different, as described in Section 3.5.1. In addition, 30-min precipitation
accumulations were used at the US site, and 60-min accumulations were used at the NOR site. It is unlikely that this biased the resultant catch efficiencies, because both 30 and 60 minute time periods are short enough that representative averages of temperature, wind speed, and precipitation type can be calculated. In addition, as described in Section 3.5.1, Wolff et al. (2015) did not detect any significant differences between 10- and 60- min intervals.

The number of events available from each site also differed, with 1156 30-min periods of single-Alter precipitation available for analysis from the US site, and only 352 60-min periods available from the NOR site. To explore the effects of a potential bias towards the more numerous US catch efficiency measurements, a test was performed using approximately one out of every four US single-Alter measurements. The resultant single-Alter dataset included 352 NOR measurements and 292 US measurements. Using the gauge height winds and the Exp. transfer function as an example, there were only small differences
between the resultant transfer function and the original transfer function created using all of the available US and NOR measurements. Likewise, the site-specific errors were not significantly altered by the omission of 3 out of every four US measurements. At the NOR site the original RMSE of 0.45 mm was unchanged by application of the new transfer function, and the bias was changed from -0.04 mm to -0.05 mm. The US results were also not significantly changed, with an increase in the RMSE from 0.15 mm to 0.17 mm, and a change in the bias from 0.00 mm to -0.01 mm. All of the available
measurements were used to develop the transfer functions presented in the main body of the results, but based on the results of this limited testing, the effects of having more US measurements than NOR measurements were not significant.

Two sites were included in this study to help advance the methods and concepts available for the development and testing of transfer functions using measurements from multiple testbeds. The focus of this work was on applying new methods to existing datasets from two sites, rather than on refining the methods used to prepare the precipitation measurements for analysis. The development of a common approach to data processing and event selection is included in WMO-SPICE. Application of the methods presented here to more standardized datasets from several WMO-SPICE sites is currently underway, and will be made available in the WMO-final report and associated publications. In addition to employing standardized methods to prepare the precipitation datasets available for transfer function development, WMO-SPICE also includes more sites with more varied climate conditions, which will be used to better quantify site-specific biases associated with the use of a single transfer function at multiple sites and create more universally applicable transfer functions.

## 6. Conclusions

Methods to address the effects of wind and precipitation type on precipitation measurements have been presented, and significant improvements to the measurements have been demonstrated for unshielded gauges and other common gauge/shield combinations. A new adjustment function was used to describe catch efficiency as a function of air temperature and wind speed, and it performed comparably to the more complex function suggested by Wolff et al. (2015). Precipitation measurements from two sites were used to derive and test a single transfer function for the single-Alter shielded measurements, and site biases caused by using one transfer function at more than one site were quantified for the first time. In addition, the remaining uncertainty in the transfer functions used to correct or standardize precipitation measurements has also been carefully described and quantified. Significant errors persisted in the measurements, even after correction for undercatch, with the RSME reduced by less than 50% for all wind shields examined. Measurement error, the random spatial variability of precipitation, and variability in the type, size, density, and fall speed of hydrometeors all likely contributed to the errors that remained uncorrected. This is an active ongoing area of research that merits more attention. However, when precipitation measurements are used to help describe water budgets, such variability in the corrected measurements may be relatively unimportant relative to the improvement in the bias, or the total amount of precipitation. In addition, this study indicates that low-porosity windshields like the Belfort double-Alter show great promise in reducing undercatch with a small-footprint, low-maintenance shield.

Significant errors exist in our historical and present-day precipitation measurements. For weighing gauges that are designed to measure snowfall, these errors are affected primarily by shielding, precipitation type, crystal habits, wet vs. dry snow, and wind speed. Such errors affect the measurement of the amount of water in both seasonal and ephemeral snowpack, and therefore affect our ability to quantify the availability of water for communities and ecosystems that rely upon water from snowfall. The results and techniques presented here can be used to help create precipitation records that are traceable to a common standard, ultimately leading to a more constrained and accurate representation of the earth's hydrological balance.

## Acknowledgements

We thank Hagop Mouradian from Environment and Climate Change Canada for contributing the mapped site locations (Fig. 1a). We also thank Samuel Buisan from the Spanish National Meteorological Agency and Eva Mekis and Michael Earle from Environment and Climate Change Canada for carefully reviewing this manuscript. This work was greatly improved by their comments.

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

**Tables and Figures**

| Shield | Uncor RMSE, mm (%) | Uncor Bias, mm (%) |
|--------|--------------------|--------------------|
| US UN | 0.30 (28.6) | -0.17 (-16.2) |
| All SA | 0.35 (34.3) | -0.16 (-16.1) |
| NOR SA | 0.64 (51.6) | -0.34 (-27.1) |
| US SA | 0.22 (23.6) | -0.11 (-11.7) |
| US DA | 0.21 (21.6) | -0.10 (-10.6) |
| US BDA | 0.16 (17.5) | -0.05 (-5.6) |
| US SDFIR | 0.14 (14.7) | -0.03 (-3.6) |

**Table 1. Errors and biases in the uncorrected 30-min precipitation from gauges under test, estimated using the DFIR precipitation measurements as the standard.**

| Shield | Sig Coef | | | | | | Exp Coef | | | $n$ | Max $U$ |
|--------|----------|----------|------------|----------|----------|----------|----------|----------|----------|-----|---------|
| | $\tau_1$ | $\tau_2$ | $T_\tau$ | $s_\tau$ | $\theta$ | $\beta$ | $a$ | $b$ | $c$ | | (m s$^{-1}$) |
| US UN | 0.31 | 0.94 | -0.08 | 0.92 | 2.58 | 1.23 | 0.063 | 1.22 | 0.66 | 843 | 6 |
| All SA | 0.20 | 0.96 | 0.22 | 1.11 | 4.70 | 1.97 | 0.040 | 1.10 | 0.54 | 1501 | 9 |
| NOR SA | 0.26 | 1.02 | 0.88 | 0.99 | 3.1 | 1.61 | 0.054 | 0.71 | 0.26 | 352 | 9 |
| US SA | 0.16 | 0.95 | -0.34 | 1.01 | 4.9 | 1.90 | 0.036 | 1.04 | 0.63 | 1156 | 6 |
| US DA | 0.00 | 0.92 | -1.19 | 1.89 | 7.04 | 1.36 | 0.028 | 0.74 | 0.66 | 1392 | 6 |
| US BDA | 0.00 | 1.00 | 1.81 | 0.57 | 8.73 | 2.87 | 0.015 | 0.32 | 0.38 | 1204 | 6 |
| US SDFIR | 0.99 | 0.96 | 0.52 | 0.10 | 0.14 | 6.16 | 0.006 | 0.00 | 0.00 | 1508 | 6 |

**Table 2. Transfer function coefficients for estimated gauge height wind speeds. Coefficients for the sigmoid transfer function (Sig, Eq. 3) and the exponential transfer function (Exp, Eq. 4), as well as the number of periods available ($n$) and the maximum wind speed (Max $U$) are described for the US unshielded (UN), single-Alter (SA), double Altar (DA), Belfort double-Alter (BDA), small DFIR (SDFIR), the NOR single-Alter (SA) gauge, and the combined US and NOR SA results (All SA). Max $U$ is included to indicate the wind speed above which the transfer function is invalid. At high wind speeds the transfer function should be applied by replacing the measured wind speed with the Max $U$.**

| Shield | Sig Coef | | | | | | Exp Coef | | | $n$ | Max $U$ |
|---|---|---|---|---|---|---|---|---|---|---|---|
| | $\tau_1$ | $\tau_2$ | $T_\tau$ | $s_\tau$ | $\theta$ | $\beta$ | $a$ | $b$ | $c$ | | (m s$^{-1}$) |
| US UN | 0.31 | 0.94 | -0.08 | 0.92 | 3.58 | 1.23 | 0.045 | 1.21 | 0.66 | 843 | 8 |
| All SA | 0.17 | 0.96 | 0.23 | 1.11 | 6.46 | 2.01 | 0.03 | 1.04 | 0.57 | 1501 | 12 |
| NOR SA | 0.25 | 1.03 | 0.95 | 1.06 | 3.99 | 1.61 | 0.05 | 0.66 | 0.23 | 352 | 12 |
| US SA | 0.12 | 0.95 | -0.35 | 1.01 | 7.05 | 1.87 | 0.03 | 1.06 | 0.63 | 1156 | 12 |
| US DA | 0.00 | 0.92 | -1.19 | 1.89 | 9.75 | 1.36 | 0.021 | 0.74 | 0.66 | 1392 | 8 |
| US BDA | 0.31 | 1.00 | 1.79 | 0.58 | 10.0 | 3.15 | 0.01 | 0.48 | 0.51 | 1204 | 8 |
| US SDFIR | 0.99 | 0.96 | 0.52 | 0.10 | 0.14 | 10.75 | 0.004 | 0.00 | 0.00 | 1508 | 8 |

**Table 3. Transfer function coefficients for 10 m height wind speeds. Coefficients for the sigmoid transfer function (Sig, Eq. 3) and the exponential transfer function (Exp, Eq. 4), as well as the number of periods available ($n$) and the maximum wind speed (Max $U$) are described for the US unshielded (UN), single-Alter (SA), double-Altar (DA), Belfort double-Alter (BDA), small DFIR (SDFIR), the NOR single-Alter (SA) gauge, and the combined US and NOR SA results (All SA). Max $U$ is included to indicate the wind speed above which the transfer function is invalid. At wind speeds greater than Max $U$, the transfer function should be applied by replacing the measured wind speed with the appropriate Max $U$ value.**

| Shield | Sig RMSE, mm (%) | Sig Bias, mm (%) | Exp RMSE, mm (%) | Exp Bias, mm (%) |
|---|---|---|---|---|
| US UN | 0.18 (16.8) | 0.00 (0.4) | 0.18 (16.8) | 0.00 (0.4) |
| All SA | 0.23 (22.3) | -0.02 (-1.5) | 0.25 (24.0) | -0.02 (-1.7) |
| NOR SA | 0.46 (36.9) | -0.06 (-4.5) | 0.47 (37.7) | -0.01 (-1.1) |
| US SA | 0.13 (14.0) | -0.01 (-0.6) | 0.14 (14.9) | -0.00 (-0.5) |
| US DA | 0.13 (13.8) | 0.00 (0.0) | 0.14 (13.8) | 0.00 (0.1) |
| US BDA | 0.12 (13.6) | -0.01 (-1.7) | 0.13 (14.6) | 0.00 (-0.9) |
| US SDFIR | 0.13 (14.2) | -0.01 (-1.6) | 0.13 (14.2) | -0.01 (-1.6) |

**Table 4. Transfer function results for estimated gauge height wind speeds. Sigmoid transfer function (Sig, Eq. 3) and exponential transfer function (Exp, Eq. 4) RMSE values and biases described for the US unshielded (UN), single-Alter (SA), double-Altar (DA), Belfort double-Alter (BDA), small DFIR (SDFIR), the NOR single-Alter (SA) gauge, and the combined US and NOR SA results (All SA).**

| Shield | Sig RMSE, mm (%) | Sig Bias, mm (%) | Exp RMSE, mm (%) | Exp Bias, mm (%) |
|---|---|---|---|---|
| US UN | 0.18 (16.8) | 0.00 (0.4) | 0.19 (17.5) | 0.00 (0.6) |
| All SA | 0.24 (22.9) | -0.02 (-1.8) | 0.25 (24.4) | -0.02 (-2.0) |
| NOR SA | 0.46 (37.3) | -0.06 (-4.5) | 0.46 (36.7) | -0.02 (-1.3) |
| US SA | 0.13 (14.0) | -0.01 (-0.6) | 0.14 (15.2) | -0.00 (-0.5) |
| US DA | 0.13 (13.9) | 0.00 (0.0) | 0.14 (13.8) | 0.00 (0.1) |
| US BDA | 0.12 (13.6) | -0.01 (-1.7) | 0.13 (14.6) | -0.00 (-0.9) |
| US SDFIR | 0.13 (14.1) | -0.01 (-0.09) | 0.13 (14.2) | -0.01 (-1.6) |

**Table 5. Transfer function results for 10 m height wind speeds. Sigmoid transfer function (Sig, Eq. 3) and exponential transfer function (Exp, Eq. 4) RMSE values and biases described for the US unshielded (UN), single-Alter (SA), double-Altar (DA), Belfort double-Alter (BDA), small DFIR (SDFIR), the NOR single-Alter (SA) gauge, and the combined US and NOR SA results (All SA).**

| Shield | Sig RMSE, mm (%) | Sig Bias, mm (%) | Exp RMSE, mm (%) | Exp Bias, mm (%) |
|---|---|---|---|---|
| NOR SA | 0.46 (36.5) | -0.06 (-4.7) | 0.45 (36.4) | -0.05 (-4.1) |
| US SA | 0.14 (14.6) | 0.00 (0.0) | 0.15 (15.4) | 0.00 ( -0.2) |

**Table 6. Transfer function results for estimated gauge height wind speeds. The same coefficients determined by the multi-site All SA transfer function (Table 2) were used to correct the results from NOR and US. Sigmoid transfer function (Sig, Eq. 3) and exponential transfer function (Exp, Eq. 4) RMSE values and biases are described for the US single-Alter (SA) and the NOR single-Alter (SA) gauge.**

| Shield | Sig RMSE, mm (%) | Sig Bias, mm (%) | Exp RMSE, mm (%) | Exp Bias, mm (%) |
|---|---|---|---|---|
| NOR SA | 0.47 (36.8) | -0.09 (-6.7) | 0.44 (34.6) | -0.09 (-7.3) |
| US SA | 0.14 (14.7) | 0.00 (0.5) | 0.15 (15.5) | 0.00 (1.0) |

**Table 7. Transfer function results for estimated 10 m height wind speeds. The same coefficients determined by the multi-site All SA transfer function (Table 3) were used to correct the results from NOR and US. Sigmoid transfer function (Sig, Eq. 3) and exponential transfer function (Exp, Eq. 4) RMSE values and biases are described for the US single-Alter (SA) and the NOR single-Alter (SA) gauge.**

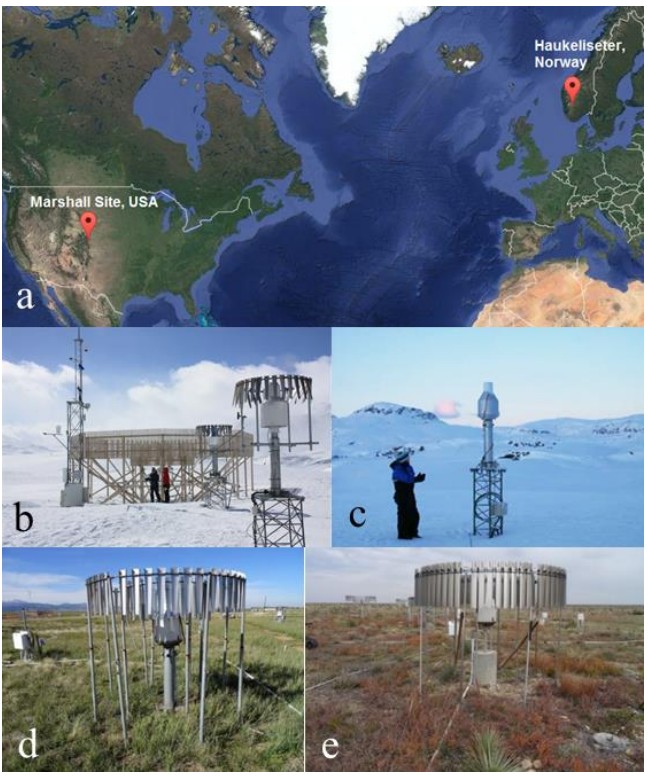

**Figure 1: (a) Site map; (b) Haukeliseter DFIR, single-Alter, and (c) unshielded gauges; (d) Marshall Belfort double-Alter and (e) double-Alter shielded gauges.**

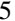

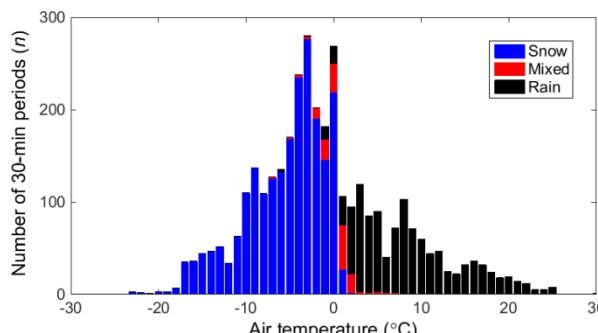

**Figure 2: The temperature distribution of 30-min periods classified as snow, mixed, and rain from the US site.**

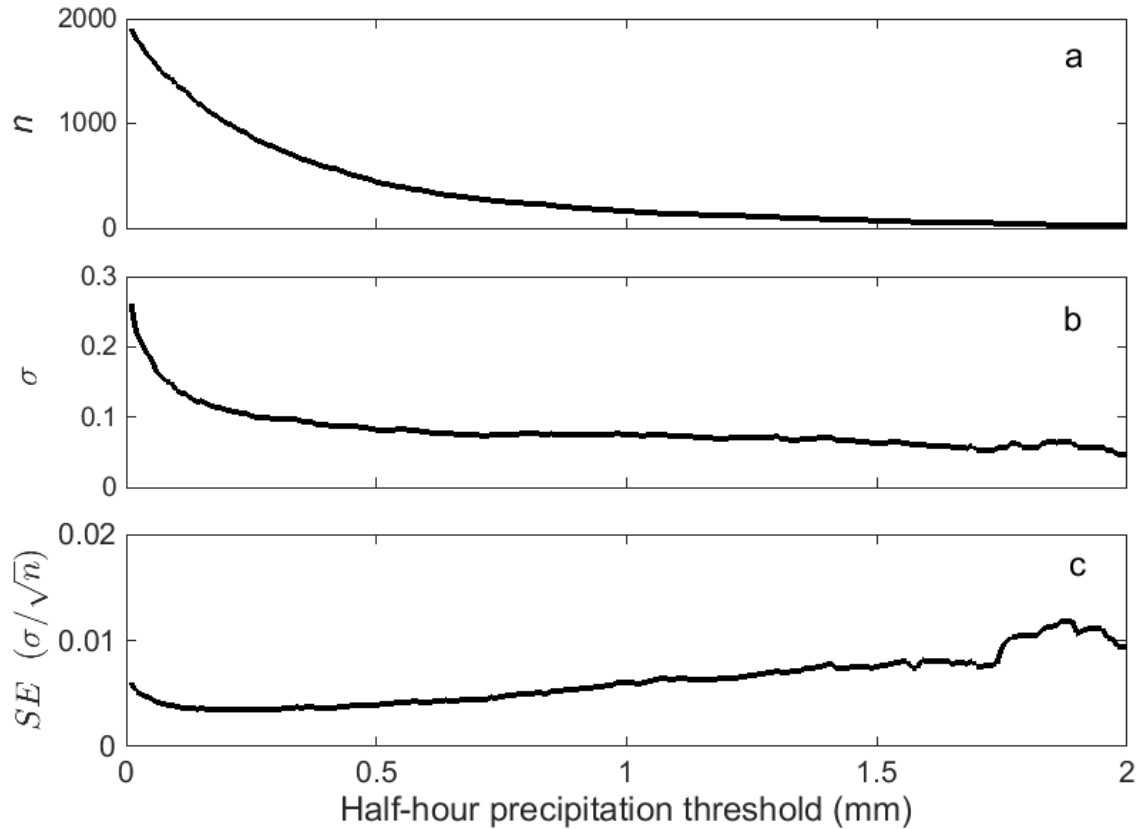

**Figure 3: The effects of minimum precipitation threshold on transfer function development for 30-min periods from the US site. (a) The number of 30-min periods ($n$) above the threshold, (b) the standard deviation ($\sigma$) of the linear transfer function error, (c) and the standard error ($SE = \sigma/\sqrt{n}$) of the transfer function are shown. Only snow results are included here, with snow in this case identified by the present weather detector with more than 15 min of snow and less than 5 min of other precipitation types within each 30-min period.**

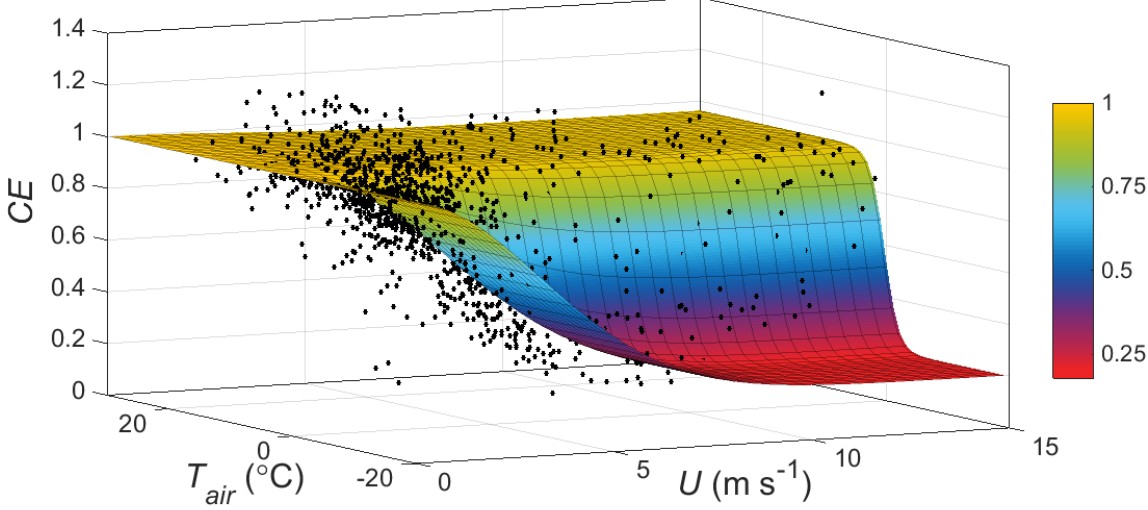

**Figure 4: Example transfer function using the sigmoid function to describe the combined SA catch efficiency (*CE*) measurements from both the US and NOR sites as a function of air temperature (*$T_{air}$*) and gauge-height wind speed (*U*). Individual 30-min *CE* measurements are shown (black circles) along with the sigmoid function fit to them (coloured surface), with the colour of the surface indicating *CE* magnitude. 352 of the measurements are from the NOR site, recorded during winter periods of 2011, 2012, and 2013, and 1156 measurements are from the US site, recorded from Jan, 2009 through March, 2014.**

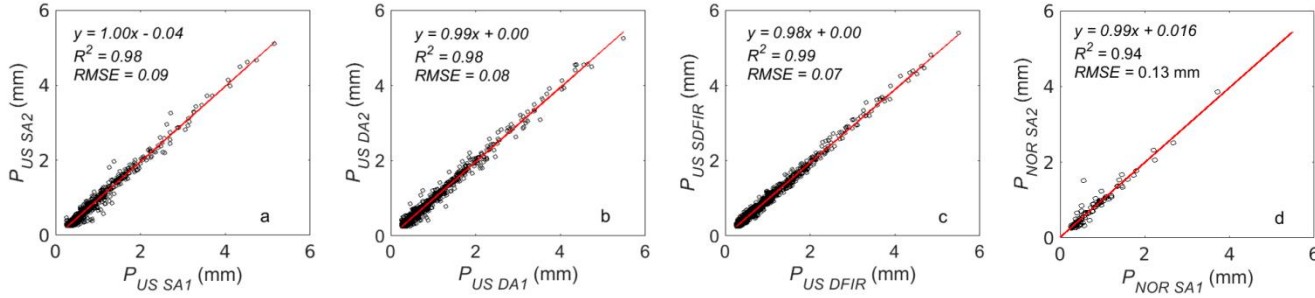

**Figure 5: Comparison of precipitation (*P*) measured from four pairs of replicate or near-replicate gauge-shield combinations. From the US site 30-min measurements recorded using (a) two single-Alter (SA) gauges, (b) two double-Alter (DA) gauges, and (c) double fence intercomparison reference (DFIR) and small DFIR (SDFIR) gauges are compared. From the NOR site, (d) 60-min measurements by two single-Alter (SA) gauges are compared. Only snow data are included here.**

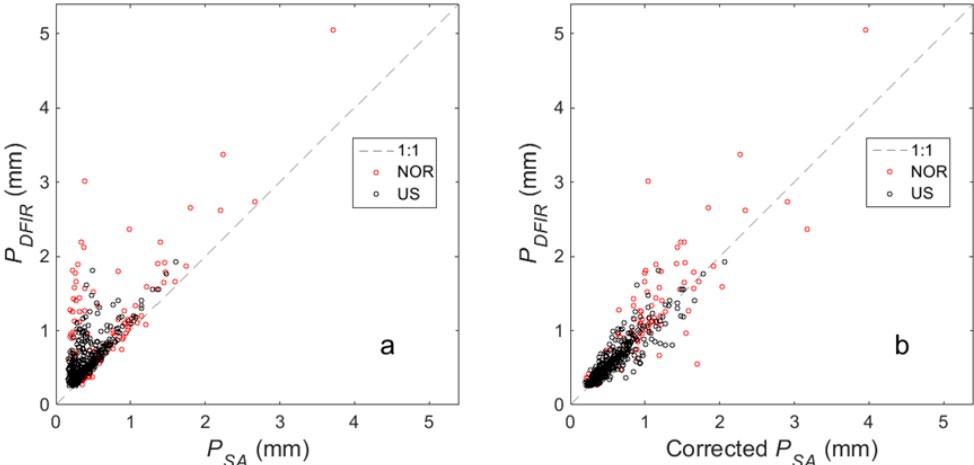

**Figure 6: – (a) Uncorrected and (b) corrected SA precipitation ($P_{SA}$) vs. DFIR precipitation ($P_{DFIR}$) for snow only, where snow is here defined as $T_{air}$ < -2.5 °C.**

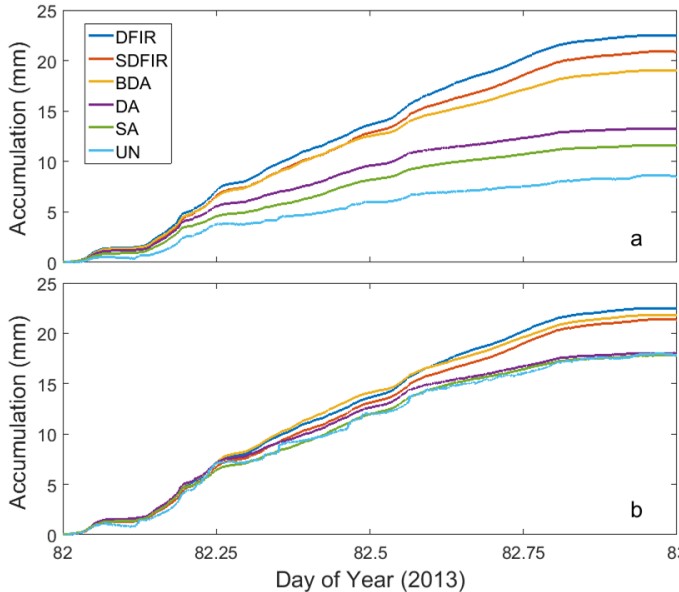

**Figure 7: Example event from the US site with accumulated precipitation measured using the Double Fence Intercomparison Reference (DFIR, dark blue), small DFIR (SDFIR, red), Belfort double-Alter (BDA, yellow), standard double-Alter (DA, purple), single-Alter (SA, green), and unshielded (UN, light blue) weighing precipitation gauges. Both the (a) uncorrected and the (b) corrected precipitation accumulations are shown, with the corrected results estimated by applying the appropriate Exp transfer**
10 **function to the 1 min accumulations. The 24 hr mean $T_{air}$ was -6.6 °C, and the mean gauge height wind speed was 3.6 m s$^{-1}$.**