# Peer review of "The Quantification and Correction of Wind-Induced Precipitation Measurement Errors"

_Hydrology and Earth System Sciences, 2016_

## Referee Comment (RC1) · E. Mekis (Referee) · 5 Oct 2016

GENERAL COMMENTS:

"The Quantification and Correction of Wind-Induced Precipitation Measurement Errors" is an important paper on the field related to addressing the quality issues of precipitation measurements using auto gauges. The paper includes observations from two sites Marshall (US) and Haukeliseter (Norway). The paper presents two transfer function models: a sigmoidal and a simplified exponential sigmoidal model, the simplified model is first published in this paper. The two models are compared using RMSE and bias statistics, the results were very similar. The paper describes the recommended / possible use of the transfer function for correcting wind related precipitation measurement errors. The comparison of different gauge / shield combination at Marshall (US)

site identifies the efficiency of different shields. The discussion on the effect using gage height versus 10 meter wind speed in the correction procedure is also important in future applications.

As of possibilities for improvements, the site descriptions should be more consistent and coordinated. The use of minimum 0.25 mm threshold is acceptable, but debatable (can be lowered, as low as 0.1 mm). For the Norwegian site further description of pre-processing method would help the understanding of the results. The time period used for the analysis is missing from the description part. The individual gauge and the combined series analysis suggested to be separated into individual tables. For the individual gauge transfer function development the US-SA and NOR-SA analysis can also be added. This additional result can also be compared with the modified coefficients using the merged ALL-SA dataset. The use of "universal" transfer function is a bit ambitious and misleading in this paper, since beside the US gauges, only one additional gauge was added to the analysis representing somewhat different climate.

SPECIFIC COMMENTS:

ABSTRACT

- It is well written. I would only argue on the use of the words "remove" in the bias analysis. In 7 out of the 8 SA verification cases (3 cases at gage height and 4 cases at 10 m wind speed) the biases was reduced (or decreased) and not removed.

CHAPTER 1

- Line 4 and 9: If possible, add more recent references here

- Line 11: Add Mekis and Vincent, 2011 reference (see at the end)

- Correct the reference: Goodison et al, 1998 (see at the end)

- Line 28: "various measuring sites" should not be used here since in the paper only two sites are included (representing two climate types only)

[Figure]

CHAPTER 2

The title "Methods" of this chapter is incorrect, misleading. It includes data (metadata) of the sites, applied gauges beside the methodology.

CHAPTER 2.1

- Add a small map with site locations

- Level of details in describing the two sites should be consistent, coordinated and synchronized: MARSHALL: Add reference gauge description; add any reference where this site is already described in details and HAUKELISETER: Add typical snow regime

- Line 12-14 sentence: Add reference to this statement

CHAPTER 2.2

- Consider adding a figure: The descriptions and all the technical details is a bit "dry" using numbers only, the comparison of the shields (for an outsider) would be easier with a sketch of the shield types

- Identify the sites (US and/or NOR) where the given shields were installed and used in this chapter (will correspond later to tables).

CHAPTER 2.3

- Line 25: Use the same notations U4.5 vs Ugh

- Identify the air temperature and wind instruments for the NOR site as well (similarly to 2.3.1), please include more details

CHAPTER 2.4

- Figure 2 reference is missing, it is probably belong to here

CHAPTER 2.5.1

- The period used for transfer function development is completely missing from the

paper (it is mentioned as "several years" in the introduction) – please include here (or somewhere in chapter 2).

- For NOR the short summary of the method from Wolff (2015) paper would be beneficial, help the understanding. Important to know how was the 3-wire input used, or any further sensor was included (wetness sensor, PWD or any other overlapping observation).

CHAPTER 2.5.2

- The use of 0.25 mm lower limit for the standard (reference) gauge is debatable based on Figure 3, the SE minimum value is around 0.1 mm.

CHAPTER 2.5.3

- Figure 4: SA data points should be better identified (period of observation, number of measurements by sites)

CHAPTER 2.6.1

- Is there any NOR pair to be included?

CHAPTER 3.2

- Table 2 and 3 are hard to read, lines are broken, difficult to find the related numbers

- The TF function development results and the validation is confused in Table 2-3. From Table 2 & 3 the US SA and NOR SA fitted values are missing, that should be part of these tables. Separate (or last line) should be used the verification values – RMSE and bias values computed from the SA-ALL coefficients. It would also add the possibility to compare the RMSE and Bias before/after values.

- Line 14: decreased the bias, not removed; it is especially true for the NOR site

- Line 25-26: easier to follow from the table, if % values are not rounded

- Lines 31-32: goes back to my original point – the algorithm at the NOR site used to

create the output from the 3 wires in missing from section 2.5.1. The difference can be due to different pre-processing algorithms as well.

- P13: Line 9: US SA N = 1156 (not 843).

- P13: Gauge height and 10 m TF analysis should be separated (in lines 6-12 the values refer to Table 2, then suddenly the conclusion is for both Table 2 and 3).

- P13 / line 15-16: Wind speed observations were available from several gauges (from 2.3.1 first paragraph), so independent gauge height wind speed measurements could have been used. Next sentence is meaningless, since the gauge height and 10m wind input are dependent (derived from each other).

- P14 / lines 1-6: Small variations are not as important as bringing the total closer to reality (in the context of the water budget) – this can be further highlighted.

CHAPTER 4.1

- Line 9-10: this statement is in contrast with the fact, that for this study the US gauge-height wind speed was derived from 10 m; in spite of the fact, that it was available at 1.5, 2 and 3 meters. The use of 10 m wind for the combined dataset is understandable, but not necessary for the individual analysis. Additional study including different wind sources could have been completed for the US installations.

CHAPTER 4.2

- As I mentioned earlier, this "universal" transfer function is not universal enough to justify the use of this word here, since the analysis representing only two different climates.

- The discussion of the site-specific analysis perhaps can be replaced by the more general climate-specific analysis.

CHAPTER 5

- Line 19: The end of the sentence was not clear, a suggestion to improve: "and for various gauge/shield combinations"

- Line 21: Verb missing at the end: Wolff et al. (2015) is presented.

REFERENCES:

- Line 16: Precipitation

- Please correct the reference: Goodison, B.E., P.Y.T. Louie, D. Yang (1998): WMO Solid Precipitation Measurement Intercomparison – Final Report. Instruments and Observing Methods Report No. 67, WMO/TD-No. 872, World Meteorological Organization, 212 p.

- Nitu et al, 2016: it was a presentation – can't be referenced.

- Add: Mekis, E. and L.A. Vincent, 2011: An overview of the second generation adjusted daily precipitation dataset for trend analysis in Canada. Atmosphere-Ocean, 49(2), 163-177.

OTHER:

- Figures should be numbered following the appearance – Figure 1 only referenced toward the end of the paper

---

## Referee Comment (RC2) · S. Buisan (Referee) · 8 Nov 2016

**GENERAL COMMENTS**

Overall this is a well written paper that should be published.

The methodology and the description of the experiment is complete and precise and can be reproduced by other scientists. The methods are clearly outlined and clearly support the conclusions. This is very important for the scope of this special issue.

My big comment is that more gauges at Norwegian site could have clearly helped in the analysis.

Another possibility would be to include in the study other sites to support the concepts of "Universal Transfer Function" and "various measuring sites". This could be the objective of another related article.

More description of Norwegian site is needed in comparison with the description of US site

SPECIFIC COMMENTS

Page 2, lines 20-21: provide a citation for "The collection of rain ... "

Page 3, line 10: Why it was "unexpected"?

Page 4, site descriptions: In NOR site you use snow depth and in US site snowfall accumulation. Please use the same variables for consistency

Page 4, site descriptions: A figure with location of the sites, pictures and layout of the sites could be useful.

Page 4, line 19: "several gauges ...." why these gauges were not included in the analysis? Problems with dataset?

Page 5, line 5: I don't understand the term "porosity"

Page 6, lines 15-17: "The 10 m wind speed .... " it seems more a conclusion than a description of the site. In addition, it is confusing to me the sentence " 10 m wind ... produced more accurate precipitations corrections ..." because later in the discussion section it is recommended to use wind speed at the gauge height.

Page 7, line 6: Where the present weather detector was located? Inside DFIR? Maybe you should reference presentation on non-catchment technologies at TECO 2016 where the limits of these instruments were presented Why data from present weather detector from NOR site was not used in the analysis? It would be really useful and also consistent to have the same Figure 2(US site) for NOR site.

Page 8, lines 19-20: please consider to describe briefly the methods for NOR site

Page 9, lines 10 -13: please refer to figure 5 to state that "SDFIR data ... was the most

comparable to DFIR" or add a citation

Page 11, lines 10 - 13: I don't understand

Page 13, lines 6 -13: Did you test the results using the same number of events for both sites? It would be a good experiment to see if the results were independent of the number of events used. Another explanation for the bias could be the wide range of wind speeds at NOR site making more difficult to obtain an accurate transfer function for NOR site.

Page 14, lines 21-23: Do you think that wind adjustment is more difficult at NOR site because of higher wind speeds and turbulence? Any study related to this? Consider more discussion and citations

Page 14, lines 25-33: Consider including that "10 m is standard according WMO guidelines and widely used in National Weather Services"

Page 14, line 31: "snow depth ... campaign ... maximum of 2m ... " it is confusing to me because previously it was written that snow depth reaches 3m

---

## Editor Comment (EC1) · M. E. Earle (Editor) · 10 Nov 2016

The paper, 'The Quantification and Correction of Wind-Induced Precipitation Errors,' by Kochendorfer et al., addresses the important hydrological topic of solid precipitation underestimation by weighing gauges used in operational networks. Errors and biases are determined for several different configurations of wind shields (relative to a reference configuration), and for gauges installed at two field sites in different climate regimes. Equations with different functional forms ('transfer functions') are tested to characterize the influences of wind speed and temperature on gauge catch efficiency, with emphasis on the uncertainty of equation outputs and the height of wind speed observations used in the equations.

This paper presents a quantitative and straightforward approach to address a complex issue, and endeavours to simplify the functional forms used in transfer functions. The application of the approach to the determination of 'universal transfer functions' is perhaps a bit lofty given the limited number of sites considered in the analysis, but the concept is novel and forward-thinking, and sets the stage well for broader implementation within the context of WMO-SPICE.

**Bigger things**

1. The paper would benefit greatly from a more detailed description of the data analysis and event selection approach used for data from the Norwegian site in Sections 2.5.1 and 2.5.2. The approach used for the US site is clearly outlined, and the selection of 30 minute event intervals is justified by the authors. Meanwhile, a reference to the approach used for the Norwegian data is provided, with no further description of the approach, and no real justification for using 60 minute intervals instead of 30 min intervals in this case. Perhaps these points are well-articulated in the reference provided, but some description of the approach is required here. This also applies to the discussion of thresholds for precipitation events – it is unclear how thresholds for the reference and weighing gauges under test were determined and implemented for the Norwegian site data/events.

2. Of greater concern are the implications of using different approaches for the different site data sets, with one approach using 30 minute time intervals, and the other using 60 minute intervals. In Sections 3.1 and 3.2, results for each site are presented and compared without mention of the caveat that the precipitation event datasets for each site were generated using different approaches that cover different time periods.

a) It would make a far stronger case for comparison if both datasets were generated in the same way, and events covered the same time interval. Is it possible to apply the same approach to data from both sites? I would suggest this as a means of strengthening the results, or at least of validating that the specific approach employed does not significantly impact the results.

Using the same time interval for events from both sites would also help to address bias observed in the results obtained using the combined transfer functions. In Section 3.2, this bias was related to the different number of precipitation events from the US and Norwegian sites that are used in the analysis. Reducing the time interval for the Norwegian events would likely increase the number of events, providing a means of testing this hypothesis.

Moreover, combined transfer functions determined using events generated in the same manner, and covering the same time interval, would provide a much firmer step toward 'universal' transfer functions.

b) Recognizing that going back and re-processing the data may be beyond the scope of this work, this paper is still an important contribution in its current state, and is suitable for publication provided the authors explicitly note the differences in how the events were generated as a complicating factor in the comparison of results from different sites, and in the determination of combined transfer functions. A discussion of the expected impacts of using different event selection approaches and time intervals on the results and comparison would also be necessary.

I feel that approach (a) will strengthen the results of this paper significantly relative to approach (b); however, approach (b) has strong 'teaser trailer potential' in terms of establishing the foundation for the work that will be done using the WMO-SPICE dataset, which includes precipitation events determined in the same way, using data that were processed in the same way, and cover the same time interval, from several different field sites/climate regimes.

**Smaller things (bold text denotes recommended additions/substitutions)**

P1, L16: 'high-quality' is a subjective qualifier, and should be removed.

P1, L20 (and throughout): 'Altar' should be replaced with 'Alter.'

P1, L28: write out 'World Meteorological Organization' and include full project acronym in parentheses, '(WMO-SPICE)'.

P2, L5-6: why are these changes in precipitation expected?

P2, L17: ….increases as wind **speed** increases.

P2, L18: collection efficiency is mentioned for the first time here, but is not defined until page 9. It needs to be defined here, perhaps just in general.

P2, L23: I don't think it's critical to note who designed the wind shields.

Section 1 (general): pictures of the different shield types should be included, and will go a long way toward clarifying the (highly detailed) descriptions in the text.

P3, L17: …a **Geonor** gauge ('Geonor' is capitalized several times throughout the paper… why?)

P3, L18-19: include ice crystal habit among factors noted.

P3, L26: 'more robust results' is another subjective qualifier… I get what you're trying to say, but I'm not sure that it's necessary.

P3, L26-32: the structure and formulation is a bit odd here. I recommend starting with 'These results include:', and then listing the different aspects.

P4, L12-14: have these results been published elsewhere?

P4, L21: the site was homogeneous in what sense?

Section 2.2: note the oil, antifreeze used for gauges at each test site.

P5, L3: the term 'porosity' should be defined.

P5, L8: here you note the lath length for the small DFIR; what is the lath length for the standard or 'tall' DFIR?

Section 2.2: here, also, some demonstrative photos would go a long way toward clarifying the text description.

Section 2.3.2: the description for the Norwegian site is significantly lacking relative to that for the US site and should be expanded (e.g. specific sensors used for ancillary measurements).

P7, L27-28: I would consider removing the part starting with ', and any temperature threshold chosen…', and reformulating the similar statement on P8, L1-2. (Otherwise, it is redundant.)

P8, L23-24: without including a demonstrative plot, the statement regarding the 'gap in spectra describing atmospheric motions' doesn't mean much to those less familiar with surface layer dynamics; I would consider removing this statement.

P9, L1-2: here you finally define catch efficiency, '…described as the ratio between…'; the ratio of what?

P9, L16-22: What do you mean by 'unbiased transfer functions'? It is also not clear to me why the test gauge being able to measure more than 0.25 mm in 30 min necessitates the use of a lower threshold for test gauges. Can you please clarify or reformulate this statement? It is important to emphasize the key role of wind shield porosity in defining the thresholds for gauges under test.

P9, L30: if you're not planning to describe Bayesian analysis in any way, I don't think it's necessary to note this here.

P10, L5: Start a new paragraph beginning with 'We also propose…'

P11, L1: I would consider changing the wording to '….to what degree a **single** transfer function…' at this point and then noting the implications for 'universal' transfer functions within the context of the results and discussion.

P11, L21: 'size' seems like an odd descriptor for the different shields (e.g. double-Alters with different slat shape/mobility are effectively the same size, but have different porosities); perhaps 'porosity' is a better term?

P11, L23: I feel like you need to justify why you would expect this, and so propose removing 'As expected' from the statement.

General note: some inconsistency with use of RMSE as singular/plural throughout. Also, I prefer 'RMSE values' to 'RMSEs' for plural use, but that's not critical.

P12, L24: change 'efficacy' of shield to 'porosity.'

Section 4.2 (P15): it is stated that the Norwegian precipitation data are noisier and that the site is much windier. Is it possible to qualify these statements with some statistics? (Mean wind speed during precip events at each site, for example.)

P15, L31: 'ephemeral' is a really great word. No comment/suggestion here, just respect.

P16, L1-2: the statement that precipitations 'must be standardized' comes off as a bit strong/preachy, and could be softened by adding '…standardized to the extent possible…'. It may not be possible to use the same approach in developing countries, for example, or different gauge types may be better suited to different climate regions.

Tables 1-3: I'm going to be blunt – your tables aren't really visually appealing. It doesn't change the content, of course, but they would look much nicer within the manuscript if they were cleaned up a bit.

Figures 1 and 7: the impact of the application of corrections to the sample dataset is mitigated by showing the corrected and uncorrected data in separate figures. I would consider changing Figure 7 to Figure 1b. Not only would this clearly demonstrate the impact of the corrections, showing this early in the paper may serve as a 'sneak preview' to entice readers to continue.

---

## Author Comment (AC1) · 7 Dec 2016

We thank the reviewers and the editor for their thorough and useful reviews.

**Response to General Comments from Eva Mekis**

As of possibilities for improvements, the site descriptions should be more consistent and coordinated. The use of minimum 0.25 mm threshold is acceptable, but debatable (can be lowered, as low as 0.1 mm). For the Norwegian site further description of pre-processing method would help the understanding of the results. The time period used for the analysis is missing from the description part. The individual gauge and the combined series analysis suggested to be separated into individual tables. For the individual gauge transfer function development the US-SA and NOR-SA analysis can also be added. This additional result can also be compared with the modified coefficients using the merged ALL-SA dataset. The use of "universal" transfer function is a bit ambitious and misleading in this paper, since beside the US gauges, only one additional gauge was added to the analysis representing somewhat different climate.

**Authors' response**: We agree with the bulk of Mekis' general comments and will change the manuscript accordingly. The site descriptions will be made more consistent, the methods used to handle the Norwegian measurements will be more thoroughly described, and the 30-min time period used for the US analysis will be described elsewhere in addition to the "Data analysis and event selection" section where it is currently described. In addition, we approve of the suggestion to create separate tables describing the combined transfer function and the individual site transfer functions and to include results within these tables describing single

Alter transfer functions developed from both sites individually. We also agree that the use of the word 'universal' overstates the additional value of using two sites, and we will change the entire manuscript accordingly.

Mekis rightly commented that the use of a 0.25 mm minimum threshold is somewhat arbitrary. For example, another statistic could be used instead of the standard error to select this threshold. In addition, the threshold determined using the standard error is sensitive to both the size of the dataset being analyzed and the amount of noise in the measurements. The resultant threshold may therefore change with the length of the measurement campaign, the frequency of precipitation, the site, the gauge, and the shield. However for the sake of consistency, in the current manuscript the same threshold was selected for all the gauges evaluated. It is also true that the standard error for a threshold of 0.1 mm was quite similar to a threshold of 0.25 mm, and it is difficult to discern a significant change in the standard error in this region on Figure 3 of the manuscript under discussion. However as shown in the modified Figure 1R (included at the end of this document), where the scale of Panel c of Figure 3 has been altered to better resolve the minimum in the standard error, the standard error was indeed lower at 0.25 mm than at 0.1 mm.

### Response to Specific Comments from Eva Mekis

ABSTRACT

- It is well written. I would only argue on the use of the words "remove" in the bias analysis. In 7 out of the 8 SA verification cases (3 cases at gage height and 4 cases at 10 m wind speed) the biases was reduced (or decreased) and not removed.

**Authors' response:** Thank you. We agree with the comments from Mekis, and the abstract will be reworded accordingly.

CHAPTER 1
- Line 4 and 9: If possible, add more recent references here
- Line 11: Add Mekis and Vincent, 2011 reference (see at the end)
- Correct the reference: Goodison et al, 1998 (see at the end)

**Authors' response:** We will add more recent references, add Mekis and Vincent, 2011, and change the year in Goodinson et al, 1997 to 1998.

- Line 28: "various measuring sites" should not be used here since in the paper only two sites are included (representing two climate types only)

**Authors' response:** We will change this to "two different sites".

CHAPTER 2
The title "Methods" of this chapter is incorrect, misleading. It includes data (metadata) of the sites, applied gauges beside the methodology.

**Authors' response:** We will add a separate chapter called, "Site Descriptions", but we believe that the section describing precipitation gauges and shields should remain within "Methods", because the measurements recorded and the gauges used can be considered part of the methods.

CHAPTER 2.1
- Add a small map with site locations

**Authors' response:** A map will be added.

- Level of details in describing the two sites should be consistent, coordinated and synchronized: MARSHALL: Add reference gauge description; add any reference where this site is already described in details and HAUKELISETER: Add typical snow regime

**Authors' response:** We will make the description of the two sites more consistent.

- Line 12-14 sentence: Add reference to this statement

**Authors' response:** A reference to Figure 5 will be added to describe the results regarding replica gauges. The analysis supporting the statement regarding the effects of wind direction on catch efficiency was never published, so we can only reference it as, "unpublished". However we can briefly summarize the results of this analysis.

CHAPTER 2.2

- Consider adding a figure: The descriptions and all the technical details is a bit "dry" using numbers only, the comparison of the shields (for an outsider) would be easier with a sketch of the shield types.

**Authors' response:** Thank you. With the same intent, in the third review the editor Michael Earle asked us to include photos of the different shields below, so we will probably do this instead of including sketches, which have already been published in Rasmussen et al. (2012). In addition, a reference to Rasmussen et al. (2012) will be added.

- Identify the sites (US and/or NOR) where the given shields were installed and used in this chapter (will correspond later to tables).

**Authors' response:** Thank you. This is a good suggestion, which will be easily accommodated.

CHAPTER 2.3

- Line 25: Use the same notations U4.5 vs Ugh

**Authors' response**: We will replace $U_{gh}$ with $U_{4.5}$.

- Identify the air temperature and wind instruments for the NOR site as well (similarly to 2.3.1), please include more details

**Authors' response:** We will include more details describing the measurements from the NOR site.

CHAPTER 2.4

- Figure 2 reference is missing, it is probably belong to here

**Authors' response:** Thank you! We will add a Figure 2 reference.

CHAPTER 2.5.1

- The period used for transfer function development is completely missing from the C3 paper (it is mentioned as "several years" in the introduction) – please include here (or somewhere in chapter 2).

**Authors' response:** Thank you. We will add the time periods describing measurements from both sites.

- For NOR the short summary of the method from Wolff (2015) paper would be beneficial, help the understanding. Important to know how was the 3-wire input used, or any further sensor was included (wetness sensor, PWD or any other overlapping observation).

**Authors' response**: Thank you. We will add a summary of Wolff et al.'s (2015) analysis methods.

CHAPTER 2.5.2

- The use of 0.25 mm lower limit for the standard (reference) gauge is debatable based on Figure 3, the SE minimum value is around 0.1 mm.

**Authors' response**: See Figure 1R near the end of this document, and also the response above to the general comments from Mekis.

CHAPTER 2.5.3

- Figure 4: SA data points should be better identified (period of observation, number of measurements by sites)
**Authors' response**: The figure caption will be augmented to include the number of measurements per site and the periods of observation.

CHAPTER 2.6.1

- Is there any NOR pair to be included?
**Authors' response**: We'll include a similar comparison plot for a pair of single Alter gauges from Norway. Figure 2R below shows the available measurements. There were some issues that affected the comparison of the two single Alter gauges, but we can describe these in more detail in the revised manuscript.

CHAPTER 3.2
- Table 2 and 3 are hard to read, lines are broken, difficult to find the related numbers
**Authors' response**: We will improve the tables. In order to fit all the values on one line and make the tables easier to read, the transfer function coefficients and the resultant RMSE and biases will be separated into two separate tables. Page breaks will also be added before the tables where necessary.

- The TF function development results and the validation is confused in Table 2-3. From Table 2 & 3 the US SA and NOR SA fitted values are missing, that should be part of these tables. Separate (or last line) should be used the verification values – RMSE and bias values computed from the SA-ALL coefficients. It would also add the possibility to compare the RMSE and Bias before/after values.

**Authors' response**: We will include the transfer function coefficients and the resultant RMSE and biases in separate tables. We will also produce site-specific transfer functions and include separate tables for the combined and individual site results.

- Line 14: decreased the bias, not removed; it is especially true for the NOR site

**Authors' response**: Thank you. We will change the wording accordingly.

- Line 25-26: easier to follow from the table, if % values are not rounded
**Authors' response**: Thank you. We weren't actually rounding, but describing the general results rather than the results of a specific correction type. These numbers weren't directly available in the tables, as they were the average of different results presented in the tables. We can clear this up by choosing a specific transfer function as an example, rather than trying to summarize the results of all the different correction types (eg. $U_{gh}$, $U_{10m}$, Sig, Exp).

- Lines 31-32: goes back to my original point – the algorithm at the NOR site used to create the output from the 3 wires in missing from section 2.5.1. The difference can be due to different pre-processing algorithms as well.
**Authors' response**: We will describe the pre-processing algorithms in the methods section.

- P13: Line 9: US SA N = 1156 (not 843).
**Authors' response**: Thank you. We will correct this.

- P13: Gauge height and 10 m TF analysis should be separated (in lines 6-12 the values refer to Table 2, then suddenly the conclusion is for both Table 2 and 3).
**Authors' response**: Thank you. We will change accordingly.

- P13 / line 15-16: Wind speed observations were available from several gauges (from 2.3.1 first paragraph), so independent gauge height wind speed measurements could have been used. Next sentence is meaningless, since the gauge height and 10m wind input are dependent (derived from each other).
**Authors' response**: Mekis rightly points out that the gauge height wind speeds were not actually measured at gauge height. For the transfer functions derived from only one site it is therefore true that similarities between the resultant gauge height and 10 m wind speed corrections are meaningless. We noted this on page 13, Ln 15-16.
However, at the NOR and US sites the relationships between the gauge height wind speed and the 10 m wind
speed were significantly different from each other; the gauge height wind speed at the NOR site was estimated as 93% of the 10 m wind speed and the gauge height at the US site was estimated as 72% of the 10 m wind speed. Assuming that catch efficiency is more closely associated with the wind speed at gauge height, one would therefore expect the errors and biases to be more significant when using the 10 m winds to determine catch efficiency at two such sites. We could not determine this using the actual gauge height wind speeds for
the transfer function development because differences in samples sizes caused by the limited uncompromised gauge height wind speed measurements actually made it more difficult to discern the effects of the two different measurement heights. We used the admittedly somewhat synthetic gauge height wind speed measurements because we wanted prove unequivocally that wind speed height affects the application of such corrections. We maintain that it is notable that we were unable to prove this.

- P14 / lines 1-6: Small variations are not as important as bringing the total closer to reality (in the context of the water budget) – this can be further highlighted.
**Authors' response**: Thank you. We will change accordingly.

CHAPTER 4.1
- Line 9-10: this statement is in contrast with the fact, that for this study the US gauge height wind speed was derived from 10 m; in spite of the fact, that it was available at 1.5, 2 and 3 meters. The use of 10 m wind for the combined dataset is understandable, but not necessary for the individual analysis. Additional study including different wind sources could have been completed for the US installations.

**Authors' response**: The discussion that Mekis objects to is based mainly on first principles, rather than our measurements. We will clarify this in the revised manuscript. Or we can certainly remove this section if the editor requests, but despite the fact that we did not use the actual gauge height wind speeds we feel we can nevertheless recommend using "the gauge height wind speed when a gauge height wind speed is available or an approximation of the gauge height wind speed". The reality is that at the NOR and US sites, for the reasons discussed above in response to Mekis' comment on P13 / line 15-16, use of the actual gauge height and near gauge height wind speed measurements to correct precipitation measurements resulted in larger errors than use of the 10 m height winds. This may in fact be a common problem due to the challenges of recording unobstructed gauge height wind speeds, but the problem is certainly not universal, nor do we believe that the actual US or NOR gauge-height wind speed measurements should be used to draw general conclusions about the accuracy of transfer functions developed using the actual gauge height wind speeds.

Additional studies were performed on the use of the different wind speed measurements, and the approach we adopted was as a result of these studies. We actually spent quite a bit of time contemplating this issue and testing different methods to evaluate the effects of different wind speed measurement heights on the resultant transfer functions. A large part of the reason we adopted the approach developed here is because we believed it would be more widely applicable, with 10 m height winds more widely available and often more representative of a monitoring site as a whole than gauge height wind speeds. The approach we developed demonstrates a practical and defensible method for adjusting the 10 m wind speeds down to the gauge height to correct precipitation measurements, and we developed this approach both to test the sensitivity of the corrections to different gauge heights and because we hoped the technique will be useful to others.

We will augment the discussion of wind speed height to clarify this.

CHAPTER 4.2

- As I mentioned earlier, this "universal" transfer function is not universal enough to justify the use of this word here, since the analysis representing only two different climates.

**Authors' response**: Thank you. We will change this here and throughout the text.

- The discussion of the site-specific analysis perhaps can be replaced by the more general climate-specific analysis.

**Authors' response**: Thank you. We will change accordingly, rewording this section to focus more on climate than site.

CHAPTER 5

- Line 19: The end of the sentence was not clear, a suggestion to improve: "and for various gauge/shield combinations"
**Authors' response**: Thank you. We will change accordingly.

- Line 21: Verb missing at the end: Wolff et al. (2015) is presented.
**Authors' response**:  The active verb in this sentence is "performed", but we can still improve the way this sentence is written.

REFERENCES:
- Line 16: Precipitation
**Authors' response**: We confirmed with Mekis in a personal communication that 'precipitation' is spelled correctly, and there is no need to correct this.

- Please correct the reference: Goodison, B.E., P.Y.T. Louie, D. Yang (1998): WMO Solid Precipitation
Measurement Intercomparison – Final Report. Instruments and Observing Methods Report No. 67, WMO/TD-No. 872, World Meteorological Organization, 212 p.
**Authors' response**: Thank you. We will correct the reference.

- Nitu et al, 2016: it was a presentation – can't be referenced.
**Authors' response**: We will inquire with HESS regarding their guidelines, and remove if necessary.

- Add: Mekis, E. and L.A. Vincent, 2011: An overview of the second generation adjusted daily precipitation dataset for trend analysis in Canada. Atmosphere-Ocean, 49(2), 163-177.
**Authors' response**: Thank you. We will change accordingly.

OTHER:
- Figures should be numbered following the appearance – Figure 1 only referenced toward the end of the paper
**Authors' response**: Thank you. We will change accordingly, either by referencing Figure 1 earlier in the manscript or by changing the order of the figures.

**Response to General Comments from Samuel Buisan**

My big comment is that more gauges at Norwegian site could have clearly helped in the analysis. Another possibility would be to include in the study other sites to support the concepts of "Universal Transfer Function" and "various measuring sites". This could be the objective of another related article.
**Authors' response**: Thank you. It is certainly true that more gauges at the Norwegian site would have been useful and that additional sites would help make the results more universal. The work presented here was performed before the WMO-SPICE measurements were available, and similar techniques will be used on the

WMO-SPICE measurements in subsequent publications. These will include more sites and more gauges. While the measurements presented in this manuscript would certainly benefit from the inclusion of more gauges and sites, the work nevertheless demonstrates new techniques (such as the Exp. function), new gauge types (such as the double Alters and the SDFIR), and for one shield type combines automated measurements to create a two-site transfer function for the first time.

More description of Norwegian site is needed in comparison with the description of US site.
**Authors' response**: Thank you. Mekis provided the same recommendation in her review, and we will augment the manuscript accordingly.

**Response to Specific Comments from Samuel Buisan**

Page 2, lines 20-21: provide a citation for "The collection of rain…"
**Authors' response**: This is a good suggestion. We will provide a citation.

Page 3, line 10: Why it was "unexpected"?
**Authors' response**: We will reword and remove the word "unexpected".

Page 4, site descriptions: In NOR site you use snow depth and in US site snowfall accumulation. Please use the same variables for consistency.
**Authors' response**: Thank you. We will change the text accordingly.

Page 4, site descriptions: A figure with location of the sites, pictures and layout of the sites could be useful.
**Authors' response**: Thank you. Mekis also suggested that we include a map. We will do this, and include example photos of the shields as well.

Page 4, line 19: "several gauges…" why these gauges were not included in the analysis? Problems with dataset?
**Authors' response**: This is a good question. Yes, the other two single Alter gauges and an unshielded gauge at NOR were too noisy to use. These problems were resolved eventually, but the measurements recorded within the time period included in the manuscript were too limited to be very useful. In addition, the different positions of the SA-gauges relative to the DFIR required the filtering of very different wind direction angles to allow for uninfluenced measurements – in total there were too few episodes to compare directly. This is apparent in the new panel (Figure 2R, included below for reference) that will be added to Figure 5 in response to one of Mekis' comments. Only 103 60-min periods of precipitation were available for comparison of the two gauges.

Page 5, line 5: I don't understand the term "porosity".

**Authors' response**: Porosity describes how porous something is, or the percent of a material that consists of holes. For a wind shield, porosity is the percent of the shield that is open, allowing air to pass through; air can pass though 50% of a DFIR fence (the other 50% is blocked by wood). This word is fairly common in English, and trying to describe the concept without it might cause confusion, but we will include more explanation in the text.

Page 6, lines 15-17: "The 10 m wind speed" it seems more a conclusion than a description of the site. In addition, it is confusing to me the sentence " 10 m wind produced more accurate precipitations corrections" because later in the discussion section it is recommended to use wind speed at the gauge height.

**Authors' response**: Thank you. This is indeed confusing, and we will clarify in the manuscript. Also please refer to the explanation of this topic included in the responses to Mekis' comments. In the methods section we must explain how the wind speeds were used to estimate the gauge height wind speeds, so we maintain that it is also appropriate to describe the reason for adopting this approach within the methods section. However we will clarify that the approach was adopted based on "preliminary analysis". In addition, the section describing our recommended use of the gauge height winds will be altered to explain more clearly that the recommendation was based solely on first principles and differences in gauge height or changes in snow depth, rather than our actual measurements. The Wind Speed for Transfer Functions discussion will also be expanded to include a more balanced description of the difficulties of recording accurate gauge height wind speeds, the advantages of estimating the gauge height winds using the 10 m winds, and the fact that our measurements did not actually indicate any advantages to using the estimated gauge height winds over the 10 m height winds.

Page 7, line 6: Where the present weather detector was located? Inside DFIR? Maybe you should reference presentation on non-catchment technologies at TECO 2016 where the limits of these instruments were presented. Why data from present weather detector from NOR site was not used in the analysis? It would be really useful and also consistent to have the same Figure 2 (US site) for NOR site.
**Authors' response**: The present weather detector was located in the open. This detail will be added to the manuscript. We will also add a reference describing the limits of these sensors, but the TECO presentation didn't actually include weather type results, as it was focused on precipitation intensity. The NOR weather type analysis was previously included in Wolff et al. (2015), and we will add this reference to the manuscript.

Page 8, lines 19-20: please consider to describe briefly the methods for NOR site.
**Authors' response**: Thank you. We will summarize the NOR event selection methods here.

Page 9, lines 10 -13: please refer to figure 5 to state that "SDFIR data was the most comparable to DFIR" or add a citation.
**Authors' response**: Thank you. We will add a reference to Table 1.

Page 11, lines 10 – 13: I don't understand

**Authors' response**: The 10-fold cross validation required iterative fitting of the transfer function to different sub-selections of the available measurements; we fit the sigmoid function to 90% of the available SDFIR measurements ten different times. Because the SDFIR catch efficiency was fairly linearly related to wind speed, it was difficult to obtain a fit using the sigmoid function; the curve fitting software frequently failed to converge
on a single SDFIR curve. It was possible to constrain the sigmoid coefficients within certain bounds and thereby produce a fit, but the constraints that worked on one set of SDFIR measurements didn't always work on another. Because of this we simply used one sigmoid curve fit to the entire population of measurements, and we validated the fit on the same population. It is important to note that this deviation from our preferred method is not significant; the error estimates produced using cross-validation were typically very similar to the
error estimates produced by circularly fitting and testing the function on the full population of measurements.

Page 13, lines 6 -13: Did you test the results using the same number of events for both sites? It would be a good experiment to see if the results were independent of the number of events used. Another explanation for the bias could be the wide range of wind speeds at NOR site making more difficult to obtain an accurate transfer
function for NOR site.
**Authors' response**: In response to this comment we performed a quick test by using only one out of every four US single Alter measurements. The resultant single Alter dataset included 352 NOR measurements and 292 US measurements.  Using the gauge height winds and the Exp. transfer function as an example, there were only small differences between the resultant transfer function and the original transfer function created using all of
the available US  and NOR measurements. Likewise the site-specific errors were not changed by the omission of 3 out of every four US measurements. The NOR results were almost identical, with a change in the RMSE from 0.45 mm to 0.45 mm and a change in the bias from -0.04 mm to -0.05 mm. The US results were also not significantly changed, with an increase in the RMSE from 0.15 mm to 0.17 mm, and a change in the bias from 0.00 mm to -0.01 mm.  The authors suggest that such differences are not significant. We also believe that
performing such an analysis and documenting it within the manuscript would not significantly increase the usefulness of the manuscript.

However as a result of this extra analysis prompted by Buisan's suggestion, the sentence suggesting that differences in the number of events affected the results will be removed.

Considering the results of this new analysis, Buisan is correct that the high wind speeds at the NOR site likely played a more important role in the magnitude of the transfer function errors than the number of available measurements. However even at low wind speeds the differences between the corrected NOR measurements and the DFIR-shielded measurements were larger from NOR than from US.

Page 14, lines 21-23: Do you think that wind adjustment is more difficult at NOR site because of higher wind speeds and turbulence? Any study related to this? Consider more discussion and citations

**Authors' response**: It was indeed more difficult to correct the NOR site because the winds were higher there. We will include some discussion of this in the manuscript.

Page 14, lines 25-33: Consider including that "10 m is standard according WMO guidelines and widely used in National Weather Services"

**Authors' response**: Thank you. We will include this.

Page 14, line 31: "snow depth… campaign maximum of 2 m…" it is confusing to me because previously it was written that snow depth reaches 3 m.

**Authors' response**: The maximum snow depth during the period studied in this paper (Feb2011-Apr2013) was indeed 1.97 m as written in the commented line. However, in other years snow depths of 3 m are often reached. Because of this, the precipitation gauges at NOR were mounted at the unusual height of 4.5 m. This is mentioned in Section 2.1 (Site Description, page 4, line 23), but we will further clarify this in Section 2.1.

**Response to General Comments from Michael Earle**

The application of the approach to the determination of 'universal transfer functions' is perhaps a bit lofty given the limited number of sites considered in the analysis, but the concept is novel and forward-thinking, and sets the stage well for broader implementation within the context of WMO-SPICE.

**Authors' response:** Thank you, and we will remove the 'universal' descriptor we over-ambitiously used to describe the transfer function.

**Bigger things**

1. The paper would benefit greatly from a more detailed description of the data analysis and event selection approach used for data from the Norwegian site in Sections 2.5.1 and 2.5.2. The approach used for the US site is clearly outlined, and the selection of 30 minute event intervals is justified by the authors. Meanwhile, a reference to the approach used for the Norwegian data is provided, with no further description of the approach, and no real justification for using 60 minute intervals instead of 30 min intervals in this case. Perhaps these points are well-articulated in the reference provided, but some description of the approach is required here. This also applies to the discussion of thresholds for precipitation events – it is unclear how thresholds for the reference and weighing gauges under test were determined and implemented for the Norwegian site data/events.

**Authors' response:** Thank you. We will add more description of the methods used on the Norwegian site.

2. Of greater concern are the implications of using different approaches for the different site data sets, with one approach using 30 minute time intervals, and the other using 60 minute intervals. In Sections 3.1 and 3.2, results for each site are presented and compared without mention of the caveat that the precipitation event datasets for each site were generated using different approaches that cover different time periods. a) It would make a far stronger case for comparison if both datasets were generated in the same way, and events covered the same time interval. Is it possible to apply the same approach to data from both sites? I would suggest this as a means of strengthening the results, or at least of validating that the specific approach employed does not significantly impact the results. Using the same time interval for events from both sites would also help to address bias observed in the results obtained using the combined transfer functions. In Section 3.2, this bias was related to the different number of precipitation events from the US and Norwegian sites that are used in the analysis. Reducing the time interval for the Norwegian events would likely increase the number of events, providing a means of testing this hypothesis. Moreover, combined transfer functions determined using events generated in the same manner, and covering the same time interval, would provide a much firmer step toward

'universal' transfer functions. b) Recognizing that going back and re-processing the data may be beyond the scope of this work, this paper is still an important contribution in its current state, and is suitable for publication provided the authors explicitly note the differences in how the events were generated as a complicating factor in the comparison of results from different sites, and in the determination of combined transfer functions. A discussion of the expected impacts of using different event selection approaches and time intervals on the results and comparison would also be necessary. I feel that approach (a) will strengthen the results of this paper significantly relative to approach (b); however, approach (b) has strong 'teaser trailer potential' in terms of establishing the foundation for the work that will be done using the WMO-SPICE dataset, which includes precipitation events determined in the same way, using data that were processed in the same way, and cover the same time interval, from several different field sites/climate regimes.

**Authors' response:** For this study, already pre-existing data sets from two different sites were used.  The NOR-dataset was prepared for the study by Wolff et al. (2015) in which a transfer function based on measurements from only one site was derived.  Wolff et al. (2015) describe the generation of 10 min and 1 h datasets, and a qualitative comparison of those NOR datasets did not reveal any significant differences. The more detailed Wolff et al. (2015) analysis was then performed on the 1 h data set because that time interval was similar to the operational measurement frequency in Norway. Both 30 and 60 minute time periods are short enough that representative averages of temperature, wind speed, and precipitation type can be calculated, and there is no other reason to believe that the choice of time interval would bias the resultant catch efficiencies. Based on the findings of Wolff et al. (2015) and also first principals, we did not expect significant differences between the 30 and 60 minute catch efficiencies.

In addition, as described in the abstract, the current study is meant as a conceptual study for the development and test of a transfer function based on datasets from multiple sites. The authors therefore focused on developing, presenting, and discussing their methods by applying them to existing datasets from two sites. The development of a common approach to data processing data and event selection is a major task within WMO-

SPICE.  Application of the methods presented in the current manuscript to more standardized datasets from several WMO-SPICE sites is currently underway, and these results will be published in the WMO-final report and associated publications.  We will add some clarification regarding these points to Sections 2.5.2 and 4.2.

**Also from Michael Earle: Smaller things (bold text denotes recommended additions/substitutions)**

P1, L16: 'high-quality' is a subjective qualifier, and should be removed.
**Authors' response**: "high-quality" will be replaced with "all-weather".

P1, L20 (and throughout): 'Altar' should be replaced with 'Alter.'
**Authors' response**: Thank you!

P1, L28: write out 'World Meteorological Organization' and include full project acronym in parentheses,
'(WMO-SPICE)'.
**Authors' response**: Thank you. We will correct this.

P2, L5-6: why are these changes in precipitation expected?
**Authors' response**: Such changes are predicted by climate models. We will change "expected" to "predicted"
and reference accordingly.

P2, L17: ….increases as wind **speed** increases.
**Authors' response**: Thank you. We will correct this.

P2, L18: collection efficiency is mentioned for the first time here, but is not defined until page 9. It needs to be
defined here, perhaps just in general.
**Authors' response**: Thank you. We will correct this.

P2, L23: I don't think it's critical to note who designed the wind shields.
**Authors' response**: The word, "scientists" will be removed.

Section 1 (general): pictures of the different shield types should be included, and will go a long way toward
clarifying the (highly detailed) descriptions in the text.
**Authors' response**: We will add photos of the different shield types.
P3, L17: …a **Geonor** gauge ('Geonor' is capitalized several times throughout the paper… why?)
**Authors' response**: Thank you. "GEONOR" will be replaced with "Geonor".

P3, L18-19: include ice crystal habit among factors noted.
**Authors' response**: We will include ice crystal habit.

P3, L26: 'more robust results' is another subjective qualifier… I get what you're trying to say, but I'm not sure
that it's necessary.

**Authors' response**: We will remove the phrase, 'more robust results'.

P3, L26-32: the structure and formulation is a bit odd here. I recommend starting with 'These results include:', and then listing the different aspects.

**Authors' response**: Thank you. We will re-write the sentence more clearly.

P4, L12-14: have these results been published elsewhere?

**Authors' response**: No, these analyses were performed on the Marshall measurements but never published. We didn't think they were significant enough to merit explicit inclusion.

P4, L21: the site was homogeneous in what sense?

**Authors' response**: We will add a more detailed description of the homogeneity of the site. Precipitation and wind measurements recorded before the DFIR shield was installed using two similar sets of instruments were in very good agreement, and indicated sufficient homogeneity of the location.

Section 2.2: note the oil, antifreeze used for gauges at each test site.

**Authors' response**: We will do this.

P5, L3: the term 'porosity' should be defined.

**Authors' response**: We will clarify what is meant by a porosity of 50%.

P5, L8: here you note the lath length for the small DFIR; what is the lath length for the standard or 'tall' DFIR?

**Authors' response**: We will include this detail.

Section 2.2: here, also, some demonstrative photos would go a long way toward clarifying the text description.

**Authors' response**: We will add some photos.

Section 2.3.2: the description for the Norwegian site is significantly lacking relative to that for the US site and should be expanded (e.g. specific sensors used for ancillary measurements).

**Authors' response**: We will augment the Norwegian site description.

P7, L27-28: I would consider removing the part starting with ', and any temperature threshold chosen…', and reformulating the similar statement on P8, L1-2. (Otherwise, it is redundant.)

**Authors' response**: Thank you! We will correct this redundancy.

P8, L23-24: without including a demonstrative plot, the statement regarding the 'gap in spectra describing atmospheric motions' doesn't mean much to those less familiar with surface layer dynamics; I would consider removing this statement.

**Authors' response**: We will remove the statement.

P9, L1-2: here you finally define catch efficiency, '…described as the ratio between…'; the ratio of what?
**Authors' response**: Thank you. We will add the word "precipitation".

P9, L16-22: What do you mean by 'unbiased transfer functions'? It is also not clear to me why the test gauge being able to measure more than 0.25 mm in 30 min necessitates the use of a lower threshold for test gauges. Can you please clarify or reformulate this statement? It is important to emphasize the key role of wind shield porosity in defining the thresholds for gauges under test.

**Authors' response**: Our goal was to develop transfer functions based on representative precipitation measurements, rather than a biased sub-selection of measurements. Because a threshold was needed for the standard and also because measurements from both the standard and the gauge under test were subject to a significant amount of random variation, this was actually somewhat challenging. To further clarify, in the example given the standard (not the gauge under test as stated by Earle in the comment above) erroneously measures more than 0.25 mm. If many such events are included in the analysis the measurements from the gauge under test will be erroneously biased low (as a result of the standard being erroneously biased high), and the resultant transfer function will therefore typically over-correct the gauge under test. Because of this, in order to handle measurements from both the DFIR and the gauge under test without biasing the results, the gauge under test also requires a minimum threshold. We will explain this better in the manuscript, and clarify by rephrasing "standard gauge/shield combination" as, "DFIR-shielded precipitation gauge".

P9, L30: if you're not planning to describe Bayesian analysis in any way, I don't think it's necessary to note this here.
**Authors' response**: The phrase, "using Bayesian analysis" will be removed.

P10, L5: Start a new paragraph beginning with 'We also propose…
**Authors' response**: A new paragraph break will be added in addition to the suggested phrase.

P11, L1: I would consider changing the wording to '….to what degree a **single** transfer function…' at this point and then noting the implications for 'universal' transfer functions within the context of the results and discussion.
**Authors' response**: Thank you. We will reword accordingly, "… to what degree a single two-site transfer function was valid for each of the two sites". This section will also require some rewording to accommodate the fact that the results will also include individual transfer functions for each of the SA sites.

P11, L21: 'size' seems like an odd descriptor for the different shields (e.g. double-Alters with different slat shape/mobility are effectively the same size, but have different porosities); perhaps 'porosity' is a better term?

**Authors' response**: With the exception of the Belfort double Alter, size was correlated with the efficacy of the wind shields. Porosity on the other hand does not describe the relationship between the SDFIR, double Altar, and single Altar catch efficiencies. We maintain that the relationship between shield size and shield efficacy is "generally" true, however we concede that it would be more accurate if we restated it as "size or efficacy".

P11, L23: I feel like you need to justify why you would expect this, and so propose removing 'As expected' from the statement.
**Authors' response**: We will remove 'As expected'.

General note: some inconsistency with use of RMSE as singular/plural throughout. Also, I prefer 'RMSE values' to 'RMSEs' for plural use, but that's not critical.
**Authors' response**: Thank you. We will check for consistency and replace RMSEs with RMSE values.

P12, L24: change 'efficacy' of shield to 'porosity.'
**Authors' response**: Thank you, but we maintain that the word 'efficacy' is appropriate here. Note that 'efficacy' is not the same word as 'efficiency'. The purpose of a wind shield is to reduce precipitation measurement errors. A shield associated with smaller errors is therefore more effective; efficacy can be determined directly by the resultant errors. Regarding porosity see above. For example, the Belfort double Alter is less porous than an SDFIR or a DFIR, but not as effective.

Section 4.2 (P15): it is stated that the Norwegian precipitation data are noisier and that the site is much windier. Is it possible to qualify these statements with some statistics? (Mean wind speed during precip events at each site, for example.)
**Authors' response**: Thank you. This is a good suggestion. We will include statistics comparing the sites such as the mean wind speeds during precipitation events, and we can also calculate errors and correlations from the single Alter gauges at both sites in similar wind speeds.

P15, L31: 'ephemeral' is a really great word. No comment/suggestion here, just respect.
**Authors' response**: Thank you!

P16, L1-2: the statement that precipitations 'must be standardized' comes off as a bit strong/preachy, and could be softened by adding '…standardized to the extent possible…'. It may not be possible to use the same approach in developing countries, for example, or different gauge types may be better suited to different climate regions.
**Authors' response**: Thank you. We will rephrase 'must be standardized' and clarify the intent of this paragraph, which was related to the way the measurements are corrected rather than how the measurements are recorded.

Tables 1-3: I'm going to be blunt – your tables aren't really visually appealing. It doesn't change the content, of course, but they would look much nicer within the manuscript if they were cleaned up a bit.

**Authors' response**: Thank you for being candid. A detailed description of the changes we will make to the tables is included in the response to Mekis. The tables will all be reformatted, and Tables 2 and 3 will each be replaced with three tables, allowing for all the results for a given shield type to be displayed on a single line.

Figures 1 and 7: the impact of the application of corrections to the sample dataset is mitigated by showing the corrected and uncorrected data in separate figures. I would consider changing Figure 7 to Figure 1b. Not only would this clearly demonstrate the impact of the corrections, showing this early in the paper may serve as a

'sneak preview' to entice readers to continue.

**Authors' response**: We agree that the large separation between Figure 1 and Figure 7 makes it difficult to compare the two, and will include Figure 7 in a separate panel of Figure 1, or alternatively make Figure 1 part of Figure 7.

[Figure]

**Figure 1R: The same plot shown in Figure 3 panel c in the manuscript under discussion, with the ranges of the x- and y-axes altered. The standard error (SE $= \sigma/\sqrt{n}$) of the transfer function is shown (black line) in addition to a horizontal line (red) placed to help locate the minimum in the standard error.**

[Figure]

Figure 2R: The same plot shown in Figure 3 panel c in the manuscript under discussion, with the ranges of the x- and y-axes altered. The standard error ($SE = \sigma/\sqrt{n}$) of the transfer function is shown (black line) in addition to a horizontal line (red) placed to help locate the minimum in the standard error.

---

## Editor Decision (ED1)

We thank the authors for their significant efforts to address the various points raised by the Reviewers and Editor, and for revising the manuscript, accordingly. Among other changes, the authors expanded sections and added new figures that improve the manuscript.

We have noted several additional points – mostly minor – to be addressed by the Authors prior to the publication of the manuscript. These points are detailed below.

1) The identification of 'site biases' when applying the combined transfer functions to data from the individual sites is an interesting and novel component of this work. In the original manuscript, the authors suggested that these biases could potentially result from the different numbers of precipitation events from each site used in the determination of the combined transfer functions. In their response to the Editor's review, the authors explained that these site biases were not likely the result of the different methods used to identify precipitation events at each site, nor the different time intervals used for events from each site. Their response to one of the reviewers (Samuel Buisan) also outlined additional testing to assess the influence of the different numbers of events on the resulting transfer functions.

   These justifications of the approach and results are important for ruling out potential factors that could contribute to the observed site biases, and we feel that they should be included in the manuscript. If the authors feel that these justifications/explanations would disrupt the flow of the manuscript, they could perhaps be added as an appendix or appendices.

2) The tables in the manuscript have been improved dramatically. To further streamline and 'declutter' these tables, we recommend that relevant units be moved to the table headers and removed from the table values. For example, change the header in Table 2 for wind speed to 'Max $U$ [m s$^{-1}$]' and remove the units from the associated table values, and do the same for the mm and % values in other tables.

3) The Introduction section would benefit from a brief introduction to the concept of transfer functions, including how they are derived and what they are used for.

4) Page 3, line 25: 'GENOR' should be changed to 'Geonor'.

5) Section 2: there is no reference to the fact that the Geonor gauges in the DFIR-shield at each site have different capacities (600 mm for US, 1000 mm for NOR). This detail should be included.

6) There is no reference to Figure 1c in the revised manuscript. This should be added.

7) The results presented in Section 3.3.1 pertaining to the height of wind speed measurements used in transfer functions appear to contradict results presented later in the manuscript. In Section 3.3.1, it is stated that 'the 10 m wind speed… produced more accurate precipitation corrections than the gauge-height wind speed.'

8) Page 12, line 2: it is stated that the transfer functions follow 'the same form presented by others.' Which others? Please provide references.

9) Page 12, line 10: aren't the NOR measurements for 60-minute intervals?

10) Page 12, line 25: for NOR, snowfall was identified 'when $T_{air}$ < -2 °C.' Is this the mean temperature over the interval, or the maximum reported temperature during the interval?

11) The manuscript refers to 'gauge height wind speed' and '10 m wind speed' extensively (e.g. page 14, line 2), but the tables indicate that these wind speeds are actually *maximum* values. If this is the case (which it appears to be), this needs to be stated explicitly in the text.

12) Page 14, line 25: the SDFIR results illustrate residual uncertainties when the wind effects are negligible relative to the gauge in the DFIR. Does it necessarily follow that uncertainties due to crystal type will also be the same as for the DFIR gauge?

13) Page 14, line 32 to page 15, line 2: improvements in results are compared for different shield configurations. RMSE and bias values are provided for the SDFIR gauge, demonstrating the minor improvement in results. It is then qualitatively stated that there were 'much more significant improvements' for other gauges, but no values are provided. This isn't a huge deal, it's just inconsistent. It is likely sufficient to make the same points using only qualitative statements and referring to the table(s).

14) Page 17, line 2: 'based on first principles…'. Which first principles? This statement is presented as an explanation, but no context is provided for its interpretation. Please clarify.

15) Page 23, caption for Table 7: '(Table 3?)' should be confirmed, and the question mark removed.

---

## Author Response (AR2)

**Reviewer comments and author responses:**

We thank the authors for their significant efforts to address the various points raised by the Reviewers and Editor, and for revising the manuscript, accordingly. Among other changes, the authors expanded sections and added new figures that improve the manuscript.

**-From the authors**: Thank you.

We have noted several additional points – mostly minor – to be addressed by the Authors prior to the publication of the manuscript. These points are detailed below.

1) The identification of 'site biases' when applying the combined transfer functions to data from the individual sites is an interesting and novel component of this work. In the original manuscript, the authors suggested that these biases could potentially result from the different numbers of precipitation events from each site used in the determination of the combined transfer functions. In their response to the Editor's review, the authors explained that these site biases were not likely the result of the different methods used to identify precipitation events at each site, nor the different time intervals used for events from each site. Their response to one of the reviewers (Samuel Buisan) also outlined additional testing to assess the influence of the different numbers of events on the resulting transfer functions. These justifications of the approach and results are important for ruling out potential factors that could contribute to the observed site biases, and we feel that they should be included in the manuscript. If the authors feel that these justifications/explanations would disrupt the flow of the manuscript, they could perhaps be added as an appendix or appendices.

**-From the authors**: A new Section 5.3 has been added to the manuscript:

**5.3 The sensitivity of site biases to differences in analysis methods**

Measurements presented in this study were recorded at two separate sites. Although similar methods were used to analyse the results from each site, the datasets available from each site were not developed exactly the same way. The methods used to determine when precipitation occurred were different, as described in Section 3.5.1. In addition, 30-min precipitation accumulations were used at the US site, and 60-min accumulations were used at the NOR site. It is unlikely that this biased the resultant catch efficiencies, because both 30 and 60 minute time periods are short enough that representative averages of temperature, wind speed, and precipitation type can be calculated. In addition, as described in Section 3.5.1, Wolff et al. (2015) did not detect any significant differences between 10- and 60- min intervals.

The number of events available from each site also differed, with 1156 30-min periods of single-Alter precipitation available for analysis from the US site, and only 352 60-min available from the NOR site. To explore the effects of a potential bias towards the more numerous US catch efficiency measurements, a test was performed using approximately one out of every four US single Alter measurements. The resultant single Alter dataset included 352 NOR measurements and 292 US

5  measurements. Using the gauge height winds and the Exp. transfer function as an example, there were only small differences between the resultant transfer function and the original transfer function created using all of the available US and NOR measurements. Likewise the site-specific errors were not significantly altered by the omission of 3 out of every four US measurements. At the NOR site the original RMSE of 0.45 mm was unchanged by application of the new transfer function, and the bias was changed from -0.04 mm to -0.05 mm. The US results were also not significantly changed, with an increase

10  in the RMSE from 0.15 mm to 0.17 mm, and a change in the bias from 0.00 mm to -0.01 mm. All of the available measurements were used to develop the transfer functions presented in the main body of the results, but based on the results of this limited testing, the effects of having more US measurements than NOR measurements were not significant.

Two sites were included in this study to help advance the methods and concepts available for the development and testing of

15  transfer functions using measurements from multiple testbeds. The focus of this work was on applying new methods to existing datasets from two sites, rather than on refining the methods used to prepare the precipitation measurements for analysis. The development of a common approach to data processing and event selection is included in WMO-SPICE. Application of the methods presented here to more standardized datasets from several WMO-SPICE sites is currently underway, and will be made available in the WMO-final report and associated publications. In addition to employing

20  standardized methods to prepare the precipitation datasets available for transfer function development, WMO-SPICE also includes more sites with more varied climate conditions, which will be used to better quantify site-specific biases associated with the use of a single transfer function at multiple sites and create more universally applicable transfer functions.

2) The tables in the manuscript have been improved dramatically. To further streamline and 'declutter' these tables, we

25  recommend that relevant units be moved to the table headers and removed from the table values. For example, change the header in Table 2 for wind speed to 'Max $U$ [m s-1]' and remove the units from the associated table values, and do the same for the mm and % values in other tables.

**-From the authors**: The tables have been changed accordingly. Below is an example.

| Shield | Uncor RMSE, mm (%) | Uncor Bias, mm (%) |
|---|---|---|
| US UN | 0.30 (28.6) | -0.17 (-16.2) |
| All SA | 0.35 (34.3) | -0.16 (-16.1) |
| NOR SA | 0.64 (51.6) | -0.34 (-27.1) |
| US SA | 0.22 (23.6) | -0.11 (-11.7) |

| | | |
|---|---|---|
| US DA | 0.21 (21.6) | -0.10 (-10.6) |
| US BDA | 0.16 (17.5) | -0.05 (-5.6) |
| US SDFIR | 0.14 (14.7) | -0.03 (-3.6) |

**Table 1. Errors and biases in the uncorrected 30-min precipitation from gauges under test, estimated using the DFIR precipitation measurements as the standard.**

3) The Introduction section would benefit from a brief introduction to the concept of transfer functions, including how they are derived and what they are used for.

-**From the authors**: The following paragraph was added to the introduction:

Adjustments, also referred to as transfer functions, are used to correct the gauge undercatch caused by the wind (e.g. Goodison, 1978). Such transfer functions are derived from precipitation testbed measurements, and describe the collection efficiency (also referred to as catch efficiency) for a specific gauge-shield system. These transfer functions determine the catch efficiency as a function of wind speed for different precipitation types (e.g. Yang et al., 1999), or more recently as a continuous function of both wind speed and air temperatures (Wolff et al., 2015).

4) Page 3, line 25: 'GENOR' should be changed to 'Geonor'.

-**From the authors**: Thank you! This has been corrected.

5) Section 2: there is no reference to the fact that the Geonor gauges in the DFIR-shield at each site have different capacities (600 mm for US, 1000 mm for NOR). This detail should be included.

-**From the authors**: The beginning of Section 3.2 has been augmented as follows: To reduce potential sources of uncertainty, all of the precipitation measurements presented here were recorded using the same model weighing precipitation gauge (3-wire T200B, Geonor Inc., Oslo, Norway), although there were differences in the Geonor gauge capacities, with 1000 mm gauges used at the NOR site and 600 mm gauges used at the US site.

6) There is no reference to Figure 1c in the revised manuscript. This should be added.

-**From the authors**: The following text was added to Section 3.2: Unshielded gauges (e.g. Fig. 1c) were present at both sites, but due to problems with the unshielded measurements from the NOR site only the unshielded measurements from the US site were used for the development and testing of transfer functions.

7) The results presented in Section 3.3.1 pertaining to the height of wind speed measurements used in transfer functions appear to contradict results presented later in the manuscript. In Section 3.3.1, it is stated that 'the 10 m wind speed… produced more accurate precipitation corrections than the gauge-height wind speed.'

**-From the authors**: This sentence in Section 3.3.1 has been removed: Based on preliminary analysis, the 10 m wind speed was more generally representative of the wind speed affecting all of the gauges throughout the site, and it produced more accurate precipitation corrections than the gauge-height wind speed.

8) Page 12, line 2: it is stated that the transfer functions follow 'the same form presented by others.' Which others? Please provide references.

**-From the authors**: Thank you. These references have been added accordingly: The form of both of these functions follows the same form presented by others (e.g. Goodison, 1978; Wolff et al., 2015; Yang et al., 1999), …

9) Page 12, line 10: aren't the NOR measurements for 60-minute intervals?

**-From the authors**: Thank you, this is a good point! The figure being referenced originally only included US measurements. The sentence now reads: "The repeatability or random error of the type of precipitation measurements used to develop the transfer functions can be estimated using four different sets of replicate gauge-shield combinations at the US and NOR testbeds."

In addition, the caption to Figure 5 was altered to accurately include the time periods used: "Figure 5: Comparison of precipitation ($P$) measured from four pairs of replicate or near-replicate gauge-shield combinations. From the US site 30-min measurements recorded using two single Alter (SA) gauges (a), two double Alter (DA) gauges (b), and double fence intercomparison reference (DFIR) and small DFIR (SDFIR) gauges (c) are compared. From the NOR site, 60-min measurements from two single Alter (SA) gauges are compared (d). Only snow data are included here."

10) Page 12, line 25: for NOR, snowfall was identified 'when $T_{air} < $ -2 °C.' Is this the mean temperature over the interval, or the maximum reported temperature during the interval?

**-From the authors**: The word, "mean" has been added to the text to specify that the mean temperature was used.

11) The manuscript refers to 'gauge height wind speed' and '10 m wind speed' extensively (e.g. page 14, line 2), but the tables indicate that these wind speeds are actually *maximum* values. If this is the case (which it appears to be), this needs to be stated explicitly in the text.

**-From the authors**: Mean wind speed was used for all the transfer function work. The table captions have been augmented with the following clarification of the maximum wind speed. "Max $U$ is included to indicate the wind speed above which the transfer function is invalid. At high wind speeds the transfer function should be applied by replacing the measured wind speed with the Max $U$."

12) Page 14, line 25: the SDFIR results illustrate residual uncertainties when the wind effects are negligible relative to the gauge in the DFIR. Does it necessarily follow that uncertainties due to crystal type will also be the same as for the DFIR gauge?

**-From the authors**: Systematic errors caused by the interaction of hydrometeors, wind, and the gauge can probably all be eliminated when comparing *corrected* SDFIR measurements to the DFIR. The text has been clarified as follows: "It is worth noting that the RMSE of even the corrected SDFIR measurements was greater than 0.1 mm, indicating that such uncorrectable errors may be due to random measurement error and site variability rather than crystal type and wind speed effects. That is, even a gauge that is shielded quite similarly to the reference, which likely responds to wind speed and crystal habit similarly to the DFIR, was subject to such errors; in a given half hour the SDFIR and DFIR shields are subject to similar hydrometeors and wind speeds, and these hydrometeors presumably behave the same way over each shield. Such uncorrectable errors can therefore be attributed to more random causes such as measurement noise and the natural spatial variability of precipitation."

13) Page 14, line 32 to page 15, line 2: improvements in results are compared for different shield configurations. RMSE and bias values are provided for the SDFIR gauge, demonstrating the minor improvement in results. It is then qualitatively stated that there were 'much more significant improvements' for other gauges, but no values are provided. This isn't a huge deal, it's just inconsistent. It is likely sufficient to make the same points using only qualitative statements and referring to the table(s).

**-From the authors**: For the sake of consistency the inclusion of specific RMSE and bias values for the SDFIR have been removed as follows: "For example, using the gauge height wind speed transfer functions, the corrected SDFIR gauge RMSE and bias shown in Table 3 were only slightly better than the uncorrected RMSE and bias shown in Table 1, whereas application of the transfer function resulted in a much more significant improvement in the unshielded and SA gauge error (UN and SA, in Tables 1, 3 and 5)."

14) Page 17, line 2: 'based on first principles…'. Which first principles? This statement is presented as an explanation, but no context is provided for its interpretation. Please clarify.

**-From the authors**: Thank you. The statement has been replaced with a description of the actual advantage of the gauge height wind speed. These changes are included here for reference: "However, based on the argument that catch efficiency is physically more closely tied to the wind speed at the height of the gauge than at a height of 10 m, the correction of precipitation measurements using a gauge height wind speed measurement, or an approximation of the gauge height wind speed, is more defensible than the use of a 10 m height wind speed.'

15) Page 23, caption for Table 7: '(Table 3?)' should be confirmed, and the question mark removed.

-**From the authors**: Thank you! In the caption "(Table 3)" has been replaced with "(Table 3)". Note that the tables have been re-numbered to follow the same order in which they are discussed in the manuscript.

**Track Changes Version of the Manuscript:**

[revised manuscript text omitted]

Figure 3: The effects of minimum threshold on transfer function development for 30-min periods from the US site. **(a)** The number of 30-min periods (*n*) above the threshold , **(b)** the standard deviation (*σ*) of the linear transfer function error , **(c)** and the standard error ($SE = \sigma/\sqrt{n}$) of the transfer function  are shown. Only snow results are included here, with snow in this case identified by the present weather detector with more than 15 min of snow and less than 5 min of other precipitation types within each 30-min period.

[Figure]

**Figure 4: Example transfer function using the sigmoid function to describe the combined SA catch efficiency (*CE*) measurements from both the US and NOR sites as a function of air temperature (*T_air*) and gauge-height wind speed (*U*). Individual 30-min *CE* measurements are shown (black circles) along with the sigmoid function fit to them (coloured surface), with the colour of the surface indicating *CE* magnitude. 352 of the measurements are from the NOR site, recorded during winter periods of 2011, 2012, and 2013, and 1156 measurements are from the US site, recorded from Jan, 2009 through March, 2014.**

[Figure]

**Figure 5: Comparison of  precipitation (*P*) measured from four pairs of replicate or near-replicate gauge-shield combinations. From the US site 30-min measurements recorded using, two single-Alter (SA) gauges (a), two double-Alter (DA) gauges (b), and double fence intercomparison reference (DFIR) and small DFIR (SDFIR) gauges (c) are compared. From the NOR site, 60-min measurements from two single-Alter (SA) gauges are compared (d). Only snow data are included here.**

[Figure]

**Figure 6: – Uncorrected (a) and corrected (b) SA precipitation ($P_{SA}$) vs. DFIR precipitation ($P_{DFIR}$) for snow only, where snow is here defined as $T_{air}$ < -2.5 °C.**

[Figure]

**Figure 7: Example event from the US site with accumulated precipitation measured using the Double Fence Intercomparison Reference (DFIR, dark blue), small DFIR (SDFIR, red), Belfort double-Alter (BDA, yellow), standard double-Alter (DA, purple), single-Alter (SA, green), and unshielded (UN, light blue) weighing precipitation gauges. Both the uncorrected (a) and the corrected (b) precipitation accumulations are shown, with the corrected results estimated by applying the appropriate Exp transfer function to the 1 min accumulations. The 24 hr mean $T_{air}$ was -6.6 °C, and the mean gauge height wind speed was 3.6 m s⁻¹.**

---

## Editor Decision (ED2)

**Proposed technical corrections for hess-2016-415 by *Kochendorfer et al.***

Page 1, line 15: replace comma after 'most important input' with semi-colon

Page 1, line 19: delete 'used' after 'Functions'

Page 1, line 21: add comma after 'In general'

Page 1, lines 21-22: remove 'corrections described as a function of air temperature and wind speed' and replace with 'the functions'

Page 1, line 25: add 'as inputs' after 'and air temperature' (and keep the comma)

Page 2, line 12: add 'in' before 'different measurement networks'

Page 3, line 19: change 'decreased with increasing wind speed' to 'decreased to varying extent with increasing wind speed'

Page 3, line 30: change 'weighing snow gauges' to 'weighing gauges'

Page 4, line 3: add comma after 'its uncertainty'

Page 4, line 3: change to 'the functional form of'

Page 4, line 4: you've already introduced transfer functions (great addition, by the way), so I think you can change '(referred to as a "transfer function")' to '(transfer function)'

Page 4, line 13: change to 'a total of ~ 200 cm snowfall'

Page 4, line 15: change 'allow' to 'allows'

Page 4, line 17: change to '…within the DFIR was used as the reference configuration for this study'

Page 4, line 23: change 'confirmed the fact' to 'indicated'

Page 5, line 16: change 'both the outer and inner tubes of the Geonor heated' to 'both the upper (exterior) and lower (interior) sections of the inlet heated'

Page 5, line 16: change 'while it drips' to 'when it drips'

Page 6, line 22: add a comma after 'NOR site'

Page 7, line 12: change 'Due to' to 'To mitigate'

Page 7, line 20: add a comma after 'At the Norwegian site'

Page 7, lines 28-29: the details regarding the relationship between the gauge height and 10 m wind speeds are condensed quickly in a list, which seems abrupt, and provides minimal context. Can the description of these details please be expanded (even slightly) for clarity?

Page 8, line 24: strange floating comma; perhaps a space after PWD21 that needs to be deleted

Page 8, line 27: change 'define' to 'describe'

Page 9, line 2: change 'aides in' to 'facilitates'

Page 9, line 4: change 'for the transfer functions developed here we use' to 'the transfer functions develop here use' (in other words, delete, 'for' and 'we')

Page 9, line 11: change to 'all of the 3-wire Geonor gauges'

Page 9, line 29: change 'following the following criteria' to 'using the following criteria'

Page 10, lines 15-17: change 'Because catch errors are best described' to 'Catch errors are described well' (line 15) and add a semi-colon after '… as the standard)' (end of line 17)

Page 10, line 18: change to 'Using all of the 30-minute periods…'

Page 11, line 3: change to 'precipitation. In other words, the DFIR gauge…'

Page 11, line 6: change to 'biased erroneously low'

Page 11, line 24: change 'which responds with an exponential decrease in CE to wind' to 'with an exponential response to wind'

Page 11, line 31: add comma after 'However'

Page 12, line 5: delete 'type of'

Page 12, line 8: add comma after 'At the US site'

Page 12, line 23: add 'measurements' after 'DFIR precipitation'

Page 13, line 15: add comma after 'At the US site'

Page 14, lines 9-10: remove comma after 'SA gauges' and add comma after 'US site'

Page 14, line 11: add comma after 'Likewise'

Page 14, line 23: change to 'in a given half hour period,'

Page 14, line 25: add comma after 'random causes'

Page 15, line 9: add comma after 'because of this'

Page 15, line 10: add comma after 'In addition'

Page 15, line 32: add 'measurements' after 'DFIR precipitation'

Page 16, line 2: add comma after 'at a given temperature'

Page 16, line 28: delete 'actually'

Page 17, line 20: add comma after 'Based on this'

Page 17, line 33: delete 'soon-to-be released' and change 'results of' to 'results from'

Page 18, line 1: remove hyphen between 'climate' and 'biases'

Page 18, line 2: change 'sites from this study' to 'sites in this study'; delete 'many'

Page 18, line 6: add 'shield' after DFIR (could be an auto gauge in a DFIR shield, not necessarily the standard DFIR)

Page 18, line 21: add 'periods' after '60-min'

Page 18, line 26: add comma after 'Likewise'

Page 19, line 14: Add 'Precipitation measurements from' before 'two sites'

Page 19, line 19: change to 'less than 50% for all wind shields examined'

Page 19, line 25: remove extra period

Page 20, line 28: change to 'precipitation data' (spelling error)

Page 20, lines 34-37: same reference repeated; please remove one

Page 21, line 17: remove repeated year

Page 21, line 31: fix 'Belgium2012, 25.'

Captions for Table 2 and Table 3:

- Add comma after '(Exp, Eq. 4)'
- Change 'double Altar (DA)' to 'double-Alter (DA)'
- Change 'small DIFR' to 'small DFIR'
- Change last line to: 'At wind speeds greater than Max $U$, the transfer function should be applied by replacing the measured wind speed with the appropriate Max $U$ value.'

Captions for Table 4 and Table 5:

- Change 'double Altar (DA)' to 'double-Alter (DA)'
- Change 'small DIFR' to 'small DFIR'

Caption for Figure 3 (first line): change to 'minimum precipitation threshold'

Caption for Figure 5 (first line): change from 'measured from' to 'measured by'

Captions for Figure 5, 6, 7: the identifiers (a, b, c, etc.) follow the descriptions, whereas the identifiers for earlier plots precede the description.

For example: (a) description vs. description (a)

---

## Author Response (AR3)

Page 1, line 15: replace comma after 'most important input' with semi-colon

**Response to editor:** These changes have been made.

Page 1, line 19: delete 'used' after 'Functions'

**Response to editor:** These changes have been made.

Page 1, line 21: add comma after 'In general'

10   **Response to editor:** These changes have been made.

Page 1, lines 21-22: remove 'corrections described as a function of air temperature and wind speed' and replace with 'the functions'

**Response to editor:** These changes have been made.

Page 1, line 25: add 'as inputs' after 'and air temperature' (and keep the comma)

**Response to editor:** These changes have been made.

Page 2, line 12: add 'in' before 'different measurement networks'

20   **Response to editor:** These changes have been made.

Page 3, line 19: change 'decreased with increasing wind speed' to 'decreased to varying extent with increasing wind speed'

**Response to editor:** These changes have been made.

25   Page 3, line 30: change 'weighing snow gauges' to 'weighing gauges'

**Response to editor:** These changes have been made.

Page 4, line 3: add comma after 'its uncertainty'

**Response to editor:** These changes have been made.

Page 4, line 3: change to 'the functional form of'

**Response to editor:** These changes have been made.

Page 4, line 4: you've already introduced transfer functions (great addition, by the way), so I think you can change '(referred to as a "transfer function")' to '(transfer function)'
**Response to editor:** These changes have been made.

5   Page 4, line 13: change to 'a total of ~ 200 cm snowfall'
**Response to editor:** These changes have been made.

Page 4, line 15: change 'allow' to 'allows'
**Response to editor:** These changes have been made.
10
Page 4, line 17: change to '…within the DFIR was used as the reference configuration for this study'
**Response to editor:** These changes have been made.

Page 4, line 23: change 'confirmed the fact' to 'indicated'
15   **Response to editor:** These changes have been made.

Page 5, line 16: change 'both the outer and inner tubes of the Geonor heated' to 'both the upper (exterior) and lower (interior) sections of the inlet heated'
**Response to editor:** These changes have been made.
20
Page 5, line 16: change 'while it drips' to 'when it drips'
**Response to editor:** These changes have been made.

Page 6, line 22: add a comma after 'NOR site'
25   **Response to editor:** These changes have been made.

Page 7, line 12: change 'Due to' to 'To mitigate'
**Response to editor:** These changes have been made.

30   Page 7, line 20: add a comma after 'At the Norwegian site'
**Response to editor:** These changes have been made.

Page 7, lines 28-29: the details regarding the relationship between the gauge height and 10 m wind speeds are condensed quickly in a list, which seems abrupt, and provides minimal context. Can the description of these details please be expanded (even slightly) for clarity?

**Response to editor:** The text has been expanded to, "To mitigate the effects of these compromised gauge height wind speed measurements, a relationship was developed to determine the gauge height wind speed using the 10 m height wind speed. The ratio was developed using unobstructed (wind direction > 240°) 10 m and gauge height wind speed measurements recorded during precipitation events. The resulting relationship was: $U_{4.5m} = 0.93 \times U_{10m}$, $R^2 = 0.99$, RMSE = 0.54 m s$^{-1}$. Using this relationship, the wind speed at 10 m was used to predict the gauge-height wind speed for all wind directions."

Page 8, line 24: strange floating comma; perhaps a space after PWD21 that needs to be deleted

**Response to editor:** These changes have been made.

Page 8, line 27: change 'define' to 'describe'

**Response to editor:** These changes have been made.

Page 9, line 2: change 'aides in' to 'facilitates'

**Response to editor:** These changes have been made.

Page 9, line 4: change 'for the transfer functions developed here we use' to 'the transfer functions develop here use' (in other words, delete, 'for' and 'we')

**Response to editor:** These changes have been made.

Page 9, line 11: change to 'all of the 3-wire Geonor gauges'

**Response to editor:** These changes have been made.

Page 9, line 29: change 'following the following criteria' to 'using the following criteria'

**Response to editor:** These changes have been made.

Page 10, lines 15-17: change 'Because catch errors are best described' to 'Catch errors are described well' (line 15) and add a semi-colon after '… as the standard)' (end of line 17)

**Response to editor:** This sentence has been rewritten as two sentences, without the suggested semicolon:

Catch errors are described well using catch efficiency, described as the ratio between precipitation accumulated in a gauge under test and the standard precipitation accumulation ($CE = {P_{UT}}/{P_{DFIR}}$, where $CE$ is catch efficiency, $P_{UT}$ is the

accumulation of precipitation from a gauge under test, and $P_{DFIR}$ is the accumulated DFIR precipitation used as the standard). Because of this, a minimum threshold is necessary to constrain errors in the denominator of the catch efficiency ratio.

Page 10, line 18: change to 'Using all of the 30-minute periods…'
5 **Response to editor:** These changes have been made.

Page 11, line 3: change to 'precipitation. In other words, the DFIR gauge…'
**Response to editor:** These changes have been made.

10 Page 11, line 6: change to 'biased erroneously low'
**Response to editor:** These changes have been made.

Page 11, line 24: change 'which responds with an exponential decrease in CE to wind' to 'with an exponential response to wind'
15 **Response to editor:** These changes have been made.

Page 11, line 31: add comma after 'However'
**Response to editor:** These changes have been made.

20 Page 12, line 5: delete 'type of'
**Response to editor:** These changes have been made.

Page 12, line 8: add comma after 'At the US site'
**Response to editor:** These changes have been made.
25
Page 12, line 23: add 'measurements' after 'DFIR precipitation'
**Response to editor:** These changes have been made.

Page 13, line 15: add comma after 'At the US site'
30 **Response to editor:** These changes have been made.

Page 14, lines 9-10: remove comma after 'SA gauges' and add comma after 'US site'
**Response to editor:** These changes have been made.

Page 14, line 11: add comma after 'Likewise'

**Response to editor:** These changes have been made.

Page 14, line 23: change to 'in a given half hour period,'

**Response to editor:** These changes have been made, with the addition of a hyphen, because 'half-hour' is now modifying 'period' ;).

Page 14, line 25: add comma after 'random causes'

**Response to editor:** These changes have been made.

Page 15, line 9: add comma after 'because of this'

**Response to editor:** These changes have been made.

Page 15, line 10: add comma after 'In addition'

**Response to editor:** These changes have been made.

Page 15, line 32: add 'measurements' after 'DFIR precipitation'

**Response to editor:** These changes have been made.

Page 16, line 2: add comma after 'at a given temperature'

**Response to editor:** These changes have been made.

Page 16, line 28: delete 'actually'

**Response to editor:** These changes have been made.

Page 17, line 20: add comma after 'Based on this'

**Response to editor:** These changes have been made.

Page 17, line 33: delete 'soon-to-be released' and change 'results of' to 'results from'

**Response to editor:** These changes have been made.

Page 18, line 1: remove hyphen between 'climate' and 'biases'

**Response to editor:** These changes have been made.

Page 18, line 2: change 'sites from this study' to 'sites in this study'; delete 'many'

**Response to editor:** These changes have been made.

Page 18, line 6: add 'shield' after DFIR (could be an auto gauge in a DFIR shield, not necessarily the standard DFIR)

**Response to editor:** These changes have been made.

Page 18, line 21: add 'periods' after '60-min'

**Response to editor:** These changes have been made.

Page 18, line 26: add comma after 'Likewise'

**Response to editor:** These changes have been made.

Page 19, line 14: Add 'Precipitation measurements from' before 'two sites'

**Response to editor:** These changes have been made.

Page 19, line 19: change to 'less than 50% for all wind shields examined'

**Response to editor:** These changes have been made.

Page 19, line 25: remove extra period

**Response to editor:** These changes have been made

Page 20, line 28: change to 'precipitation data' (spelling error)

**Response to editor:** These changes have been made.

Page 20, lines 34-37: same reference repeated; please remove one

**Response to editor:** These changes have been made.

Page 21, line 17: remove repeated year

**Response to editor:** These changes have been made. The same issue with Nitu et al. (2016) was also identified and fixed.

Page 21, line 31: fix 'Belgium2012, 25.'

**Response to editor:** These changes have been made.

Captions for Table 2 and Table 3:

- Add comma after '(Exp, Eq. 4)'
- Change 'double Altar (DA)' to 'double-Alter (DA)'
- Change 'small DIFR' to 'small DFIR'
- Change last line to: 'At wind speeds greater than Max U, the transfer function should be applied by replacing the measured wind speed with the appropriate Max U value.'

**Response to editor:** These changes have been made.

Captions for Table 4 and Table 5:
- Change 'double Altar (DA)' to 'double-Alter (DA)'
- Change 'small DIFR' to 'small DFIR'

**Response to editor:** These changes have been made.

Caption for Figure 3 (first line): change to 'minimum precipitation threshold'

**Response to editor:** These changes have been made.

Caption for Figure 5 (first line): change from 'measured from' to 'measured by'

**Response to editor:** These changes have been made.

Captions for Figure 5, 6, 7: the identifiers (a, b, c, etc.) follow the descriptions, whereas the identifiers for earlier plots precede the description. For example: (a) description vs. description (a)

**Response to editor:** This has been corrected.

Marked up version of the manuscript showing all the technical corrections:

[revised manuscript text omitted]